# On the Convergence of Cyclic Hierarchical Federated Learning with Heterogeneous Data

## Abstract

Hierarchical Federated Learning (HFL) advances the classic Federated Learning (FL) by introducing the multi-layer architecture between clients and the central server, in which edge servers aggregate models from respective clients and further send to the central server. Instead of directly uploading each update from clients for aggregation, the HFL not only reduces the communication and computational overhead but also greatly enhances the scalability of supporting a massive number of clients. When HFL operates for applications having a large-scale clients, edge servers train their models in a cyclic pattern (a ring architecture) as opposed to the star-type of architecture where each edge develops their own models independently. We refer it as Cyclic HFL(CHFL). Driven by its promising feature of handling data heterogeneity and resiliency, CHFL has a great potential to be deployed in practice. Unfortunately, the thorough convergence analysis on CHFL remains lacking, especially considering the widely-existing data heterogeneity issue among clients. To the best of our knowledge, we are the first to provide a theoretical convergence analysis for CHFL in strongly convex, general convex, and non-convex objectives. Our results demonstrate the convergence rate are $\tilde{\mathcal{O}}(1/MNRKT)$ for strongly convex objective, $\mathcal{O}(1/\sqrt{MNRKT})$ for general convex objective, and $\mathcal{O}(1/\sqrt{MNRKT})$ for non-convex objective, under standard assumptions. Here, $M$ is the number of edge servers, $N$ is the number of clients in edge, $K$ is local steps in client, and $R$ is the edge training round. Through extensive experiments on real-world datasets, besides validating our theoretical findings, we further show CHFL achieves a comparable or superior performance when accounting for both inter- and intra-edge data heterogeneity.

## 1 Introduction

Federated Learning (FL) McMahan et al. (2017) has emerged as a promising distributed learning framework in which a large number of distributed devices collaborate to train a joint model without sharing their data. Although FL has attracted a significant interest on theoretical research, the standard FL architecture does not always perform well in practical scenarios. When the number of clients (devices) participate in the FL, the communication burden in between the central server and each client for model update and aggregation may significantly impact the FL performance. As a solution, the edge-based FL has been proposed Wu et al. (2020) to replace a single central server with multiple edge servers. In particular, each edge server acts as a parameter server responsible for a smaller set of clients, who can meet specific communication or data requirements. However, as stated in Wang et al. (2021a), this edge-based FL still suffers the performance drop due to the limited number of participating clients managed by each edge.

To alleviate the communication burden on a central server, Hierarchical Federated Learning (HFL) Liu et al. (2020); Deng et al. (2021) introduces a three-layer architecture, i.e., a central server, multiple edge servers, and clients. Each edge server, along with its associated clients, forms an edge. The HFL has two levels of model updates including edge model update and global model update, in which many edge model updates follow the star architecture Lee et al. (2020) to aggregate local model updates from clients. On the other hand, most HFL works Liu et al. (2022; 2023; 2020); Khan et al. (2023) also assume the global model update follows the star architecture, i.e., the centralized HFL, and provides convergence analysis based on various assumptions. Rather than the star architecture,

for many FL applications across a large geographic regions Zhu et al. (2021); Paulik et al. (2021); Yang et al. (2018); Lee et al. (2020), the ring architecture is a better fit in terms of the participation pattern, where each edge server updates their models to another edge server rather than the central server, namely, the Cyclic HFL (CHFL). Compared to the star-like architecture with centralized HFL, CHFL improves scalability by accommodating more clients without being affected by central server dropout issues when the communication burden increases. Unfortunately, while a few recent works Li & Lyu (2024); Cho et al. (2023) have discussed the convergence rate in the cyclic pattern, they are limited to FL, and a comprehensive analysis under standard assumptions in HFL remains lacking. In this work, our contributions are as follows,

- We derive convergence guarantees for CHFL on heterogeneous data under standard assumptions for strongly convex, general convex, and non-convex objectives, all of which are compared with the state-of-the-art as in Tab. 1. We note that our convergence rates have the highly desirable speedup effect in terms of both edge server number $M$ and edge round $R$. As a result of generality, several well-studied FL variants such as Li & Lyu (2024); Karimireddy et al. (2020) become special cases of our framework, further echoing the correctness of our conclusion. Compared with other centralized HFL variants, we achieve the best convergence rate without considering the transmission latency between the central server and the edge server.

- We provide insights into achieving optimal convergence improvements by clustering clients with different objectives. Unlike current clustering policies, such as solely grouping clients with similar data Liu et al. (2020); Wang et al. (2022) or ensuring edges share similar data Mhaisen et al. (2021); Deng et al. (2021), our approach determines the best policy based on the settings of the number of edges, clients, local steps, and edge training rounds for various objectives. As in the general convex case, having all edges sharing similar data will lead to an optimal convergence improvement when the number of edges is relatively small. On the other hand, clustering clients with similar data will help achieve an optimal convergence improvement if the number of edges is large.

- We validate our findings with comprehensive simulation-based study on real-world datasets. The experimental results show that CHFL can achieve comparable or superior performance in terms of accuracy and convergence speed measured by local model updates. Meanwhile, we show that the edge training epoch accelerates the convergence speed, and the inter-edge heterogeneity has more effect on convergence speed than the intra-edge heterogeneity in specific conditions.

## 2 RELATED WORK

### 2.1 HIERARCHICAL FEDERATED LEARNING

In addressing the challenges of high communication overhead and latency in vanilla FL, HFL Liu et al. (2020); Bonawitz et al. (2019); Zhou & Cong (2019) was proposed to add a layer of edge servers, which simplifies the communication process to occur only between edge servers and central server Wang et al. (2021b). Liu *et. al* in Liu et al. (2020) prove HFL could achieve convergence amidst inter-edge data heterogeneity, yet the impact of intra-edge data heterogeneity on convergence remains unknown. Similarly, Abad *et.al* Abad et al. (2020) show the reduced communication latency by deploying HFL in a real mobile edge computing system. Xu *et al.* in Xu et al. (2021) introduce an adaptive HFL approach, focusing on optimal resource allocation and control of edge intervals to enhance training accuracy. OUEA Mhaisen et al. (2021) and SHARE Deng et al. (2021) consider the clients cluster problem in HFL. OUEA Mhaisen et al. (2021) cluster clients with similar data into one edge to improve performance in HFL but they only consider the convex objective. SHARE Deng et al. (2021) ensures each edge shares similar data to reduce data heterogeneity among edges to improve performance. Both discuss cluster policies in the context of centralized HFL, but these policies cannot be directly applied to CHFL to enhance performance. Our convergence analysis reveals that the effectiveness of cluster policies depends on system settings and their specific objectives.

Table 1: Convergence rates for FL and HFL variants.

| System | Method | Cyclic Pattern | Convexity[7] | Convergence Rate[8] |
|---|---|---|---|---|
| FL | Li & Lyu (2024)[1] | ✓ | SC | $\tilde{\mathcal{O}}\left(\frac{1}{CKT} + \frac{1}{MT^2}\right)$ |
| | | | GC | $\mathcal{O}\left(\frac{1}{CKT} + \frac{1}{(CK)^{1/3}T^{2/3}}\right)$ |
| | | | NC | $\mathcal{O}\left(\frac{1}{\sqrt{CKT}} + \frac{1}{(CK)^{1/3}T^{2/3}}\right)$ |
| | Karimireddy et al. (2020) | × | SC | $\mathcal{O}\left(\frac{1}{CKT} + \frac{1}{T^2}\right)$ |
| | Koloskova et al. (2020) | × | SC | $\mathcal{O}\left(\frac{1}{CKT} + \frac{1}{T^2}\right)$ |
| | Cho et al. (2023)[2] | ✓ | NC | $\tilde{\mathcal{O}}\left(\frac{M}{KNT} + \frac{ML}{NT}\left(\frac{MN/M-N}{MN/M-1}\right)\right)$ |
| HFL | Liu et al. (2022)[3] | × | NC | $\mathcal{O}\left(\frac{1}{\sqrt{RKTMN}} + \frac{1}{RKT}\right)$ |
| | Liu et al. (2023)[4] | × | NC | $\mathcal{O}\left(\frac{1}{\sqrt{RKT}}\right)$ |
| | Khan et al. (2023)[5] | × | SC | $\mathcal{O}\left(\frac{1}{T}\right)$ |
| | Liu et al. (2020)[6] | × | SC | $\mathcal{O}\left(\frac{1}{TG(R,K)}\right)$ |
| | **This paper** | ✓ | SC | $\mathcal{O}(\frac{1}{MNRKT})$ |
| | | | GC | $\mathcal{O}(\frac{1}{\sqrt{MNRKT}})$ |
| | | | NC | $\mathcal{O}(\frac{1}{\sqrt{MNRKT}})$ |

[1] The Sequential Federated Learning(SFL) with a full client participation.
[2] The cyclic Federated Learning on the non-convex case with Polyak-Łojasiewicz.
[3] Hier-Local-QSGD reduces the communication cost between the cental server and clients.
[4] Group-FEL with a wise client sampling strategy to improve the convergence speed.
[5] HSFL addresses the issue of limited computational resources on local devices.
[6] $G(R,K)$ represents the function containing local steps $K$ and edge round $R$.
[7] Shorthand notations: SC: Strongly Convex, GC: General Convex, NC: Non-Convex.
[8] We omit absolute constants and polylogarithmic factors.

## 2.2 CYCLIC/SEQUENTIAL FEDERATED LEARNING

In vanilla Federated Learning, clients exhibit system heterogeneity. Hence, it is advisable to select qualified devices suited for FL, ensuring they have a stable network for efficient model updates, sufficient charging to manage energy use, and idle status to avoid disruptions. Compared with the vanilla FL setting with random device selection Hard et al. (2018); Huba et al. (2022); Paulik et al. (2021), those qualified devices usually participate in FL at specific time and follow a cyclic pattern Zhu et al. (2021); Paulik et al. (2021); Yang et al. (2018); Lee et al. (2020). Cho *et al.* in Cho et al. (2023) explores various gradient update methods in FL under a cyclic pattern. However, their convergence analysis is based on the assumption that the local client's objective conforms to the Polyak-Łojasiewicz condition Karimi et al. (2016), a limitation considering that objectives in FL are often general non-convex Das et al. (2022). Li *et al.* in Li & Lyu (2024) offer the convergence analysis for both Parallel Federated Learning (PFL) and Sequential Federated Learning (SFL) with convex and non-convex objectives. They have the result that SFL has a better guarantee than PFL in specific conditions. However, they both discuss the convergence analysis on the two-layer FL, for which the cyclic pattern on HFL is still missing. To address this gap, we extend the application of the cyclic pattern to HFL and provide a convergence analysis for both convex and non-convex objectives. Furthermore, we delve into the impact of data heterogeneity on convergence speed across various client participant patterns in HFL. We attain optimal convergence rate compared to other studies Liu et al. (2023); Xu et al. (2021).

## 3 PRELIMINARY ON CYCLIC HFL

### 3.1 NOTATIONS

We consider a CHFL system having a set of edge servers (interchangeably with edges) $\mathcal{M}$. Each edge server $i \in \mathcal{M}$ will serve clients $\mathcal{N}_i$ with $N = |\mathcal{N}_i|$. For each client $j \in \mathcal{N}_i$, it has the local empirical loss function, $F_j(\mathbf{x}) = \frac{1}{|\mathcal{D}_j|}\sum_{\xi \in \mathcal{D}_j} \ell(\mathbf{x}, \xi)$, where $\mathcal{D}_j$ is the training dataset and $\ell(\mathbf{x}, \xi)$ is the loss value of the model $\mathbf{x} \in \mathbb{R}^d$ at data sample $\xi$. For each edge server $i$, it optimizes

$f_i(\mathbf{x}) := \frac{1}{|\mathcal{N}_i|} \sum_{j \in \mathcal{N}_i} F_j(\mathbf{x})$. The global optimization task is identical to that of standard FL where the global objective is $f(\mathbf{x}) := \frac{1}{M} \sum_{i \in \mathcal{M}} f_i(\mathbf{x})$ with $M = |\mathcal{M}|$ and the model can be founded by achieving $\min_{\mathbf{x} \in \mathbb{R}^d} f(\mathbf{x})$. Please refer to Appendix A for all symbol notations.

### 3.2 Assumptions

**Assumption 3.1** (Bounded variance). *For the local objective $F_j(\mathbf{x})$ in any client, the local stochastic gradient $\nabla F_j(\mathbf{x}, \xi_j)$ computed using a mini-batch $\xi_j$, sampled uniformly at random from local dataset $\mathcal{D}_j$, has bounded variance, that is $\mathbb{E}\|\nabla F_j(\mathbf{x}, \xi_j) - \nabla F_j(\mathbf{x})\| \leq \sigma^2$, for all clients.*

**Assumption 3.2** (Smoothness). *Smoothness of $F_j(\mathbf{x}), \forall j \in \mathcal{N}_i, \forall i \in \mathcal{M}$. The clients' local objective functions are all L-smooth, i.e., $\|\nabla F_j(\mathbf{x}) - \nabla F_j(\mathbf{x}')\| \leq L\|\mathbf{x} - \mathbf{x}'\|$ for all $\mathbf{x}$ and $\mathbf{x}'$.*

**Assumption 3.3** (Intra-Edge & Inter-Edge Data Heterogeneity for Convex Objectives). *There exist constants $\sigma_c, \sigma_g \geq 0$, such that for all $\mathbf{x}$, for all $i \in \mathcal{M}$ and for all $j \in \mathcal{N}_i$, $\left\|\nabla F_j(\mathbf{x}) - \frac{1}{|\mathcal{N}_i|} \sum_{j \in \mathcal{N}_i} \nabla F_j(\mathbf{x})\right\| \leq \sigma_c$, and $\left\|\frac{1}{|\mathcal{N}_i|} \sum_{j \in \mathcal{N}_i} \nabla F_j(\mathbf{x}) - \nabla f(\mathbf{x})\right\| \leq \sigma_g$.*

**Assumption 3.4** (Intra-Edge & Inter-Edge Data Heterogeneity for Non-Convex objective). *There exists two constants $\sigma_c, \sigma_g \geq 0$, such that $\frac{1}{M} \sum_{i=1}^{M} \|\nabla f_i(\mathbf{x}^*)\|^2 = \sigma_g^2$ and $\frac{1}{N} \sum_{j=1}^{N} \|\nabla F_j(\mathbf{x}^*)\|^2 = \sigma_c^2$ where $\mathbf{x}^* \in \arg\min_{\mathbf{x} \in \mathbb{R}^d} f(\mathbf{x})$ is one global minimizer.*

The first two assumptions are standard in both convex and non-convex optimization Ghadimi & Lan (2013); Bottou et al. (2018); Li & Lyu (2024); Yang et al. (2021). For Assumption 3.3, the bounded data heterogeneity is also a standard assumption in FL with different architectures Liu et al. (2020); Li & Lyu (2024); Yang et al. (2022); Li et al. (2019); Cho et al. (2023), which is used for non-convex cases. If all clients in one edge train model on Independent and Identically Distributed (IID) data, i.e. all clients in one edge share similar data and all edge may share different data, then $\sigma_c \simeq 0$. If all edges train model on IID data, i.e., all client in one edge may share different data and all edge share similar data, then $\sigma_g \simeq 0$. A larger $\sigma_c$ or $\sigma_g$ indicates a higher level of data heterogeneity. We take similar data as data with same label set and different data as data with different label set. Following Li *et al.* Li & Lyu (2024) and Koloskova *et al.* Koloskova et al. (2020), Assumption 3.4 uses one weaker assumption to bound the diversity on intra-edge and inter-edge only at the optima for the convex case.

### 3.3 Description of Cyclic HFL

We assume all edge servers participate in a natural cyclic pattern without the guidance from the central server, as shown in Fig.1. The training process of CHFL is as follows. In each global update $t = 0, ..., T - 1$, the edge servers randomly forms a participated queue, $\mathcal{Q}$, with $|\mathcal{Q}| = M$. Hence, the cyclic process is as follows, when $t = 0$, the initialized model $\mathbf{x}_0$ is randomized or pre-trained by a public dataset. This model can be used as an initialized model by the first edge server in $\mathcal{Q}$. Then, the traditional FL is run in the first edge, in which $N$ clients are selected to train their own model with $K$ steps and upload model updates to the edge server. The above process repeats for $R$ edge rounds. After that, the aggregated edge model will be sent to the next edge server in $\mathcal{Q}$ as an initialized model for training, until all selected edges finish the training process. We refer to the above process as one global round. In the next global round, the first edge server will receive an updated model from the last edge server of the previous global round. A more detailed process is shown in Alg. 1.

## 4 Convergence Theory

In this section, we conduct the convergence analysis on the strongly convex, general convex, and non-convex cases for CHFL (See proof details in Appendix C). By comparing with the convergence rate of other state-of-the-art HFL algorithms, our convergence rate is the optimal. To be more insightful for practical applications, we also present the convergence rate when adopting partial edge/client participate in the CFL.

**Theorem 4.1.** *For CHFL (Algorithm 1), with Assumptions 3.1, 3.2, 3.4 for strongly convex and general convex, Assumptions 3.1, 3.2, and 3.3 for non-convex case, set $\tilde{\eta} := MNRK\eta$, $\Pi_{sc} :=$*

Figure 1: System architecture of CHFL

**Algorithm 1** Cyclic HFL

**Initialization:** $\mathbf{x}_0$
In each global round $t \in \{0, 1, \ldots, T-1\}$,
sample an edge server permutation $\mathcal{Q}$
**for** edge server $i \in \mathcal{Q}$ **do**
 **for** *edge round* $r = 0$ to $R - 1$ **do**
  **for** client $j \in \mathcal{N}_i$ **do**
   **for** *local step* $k = 0$ to $K - 1$ **do**
    $\mathbf{x}_{r,k+1}^{i,j} = \mathbf{x}_{r,k}^{i,j} - \eta \mathbf{g}_{r,k}^{i,j}\,{}^{a}$
   **end for**
  **end for**
  $\mathbf{x}_{r+1}^i = \mathbf{x}_r^i - \frac{\eta}{|\mathcal{N}_i|} \sum_{j \in \mathcal{N}_i} \mathbf{g}_{r,j}\,{}^{b}$
 **end for**
 Transmit $\mathbf{x}^i$ to next edge server.
**end for**

${}^{a}\mathbf{g}_{r,k}^{i,j} = \nabla F_j(\mathbf{x}_{r,k}^{i,j}, \xi_{r,k}^{i,j})$
${}^{b}\mathbf{g}_{r,j} = \sum_{k=0}^{K-1} \nabla F_j(\mathbf{x}_{r,k}^{i,j}, \xi_{r,k}^{i,j})$

$\mathbb{E}\left[f\left(\mathbf{x}^T\right) - f\left(\mathbf{x}^*\right)\right]$, $\Pi_{gc} := \mathbb{E}\left[f\left(\mathbf{x}^T\right) - f\left(\mathbf{x}^*\right)\right]$ *and* $\Pi_{nc} := \min_{0 \le t \le T} \mathbb{E}\left[\left\|\nabla f\left(\mathbf{x}^{(t)}\right)\right\|^2\right]$, *we have the following upper bounds,*

**Strongly convex:** *With the following learning rate condition,* $\frac{1}{\mu T} \le \tilde{\eta} \le \frac{1}{35L}$. *We have the upper bound,*

$$\Pi_{sc} \le \underbrace{5\mu D^2 \exp\left(-\frac{\mu \tilde{\eta} T}{2}\right)}_{\text{Optimization term}} + \underbrace{\frac{27\tilde{\eta}\sigma^2}{MNRK} + \frac{18L\tilde{\eta}^2(M^2NR^2 + NR^2 + 1)\sigma_c^2}{M^2N^2K^2} + \frac{53L\sigma_g^2\tilde{\eta}^2}{M}}_{\text{Error terms}} \quad (1)$$

**General convex:** *With the following learning rate condition,* $\tilde{\eta} \le \frac{1}{35L}$. *We have the upper bound,*

$$\Pi_{gc} \le \underbrace{\frac{10D^2}{3\tilde{\eta}T}}_{\text{Optimization term}} + \underbrace{\frac{27\tilde{\eta}\sigma^2}{MNRK} + \frac{18L\tilde{\eta}^2(M^2NR^2 + NR^2 + 1)\sigma_c^2}{M^2N^2K^2} + \frac{53L\sigma_g^2\tilde{\eta}^2}{M}}_{\text{Error terms}} \quad (2)$$

**Non-convex:** *With the following learning rate condition,* $\tilde{\eta} \le \frac{1}{35L}$. *We have the upper bound,*

$$\Pi_{nc} \le \underbrace{\frac{2[f(\mathbf{x}^0) - f(\mathbf{x}^*)]}{\tilde{\eta}T}}_{\text{Optimization term}} + \underbrace{\frac{12L\tilde{\eta}\sigma^2}{5MNRK} + \frac{2L^2 q_\sigma(M, N, R, K)}{MNRK}(\sigma_g^2 + \sigma_c^2)\tilde{\eta}^2}_{\text{Error terms}} \quad (3)$$

*where* $q_\sigma(M, N, R, K) = \frac{2R + 3(R-1)(M-1)NK + 3(M-1)(K-1) + \frac{2(K-1)}{RN} + 3(K-1)}{6M} + \frac{RNK(M-1)}{3}$ *for the non-convex case and* $D := \left\|\mathbf{x}^0 - \mathbf{x}^*\right\|$ *for the convex case.*

In Theorem 4.1, we have our effective learning rate $\tilde{\eta} := MRNK\eta$, in which the edge number $M$ and edge round $R$ are induced by the hierarchical architecture. With a larger $\tilde{\eta}$, the optimization term will get vanished, while the error terms would be larger. Therefore, Corollary 4.2 can help find an appropriate $\tilde{\eta}$ to achieve a balance between two parts. Compared with the two-layer FL algorithms Li & Lyu (2024); McMahan et al. (2017), we specifically focus on the inter-edge data heterogeneity.

**Corollary 4.2.** *Applying the results of Theorem 4.1, with Assumptions 3.1, 3.2, 3.4 for strongly convex and general convex, Assumptions 3.1, 3.2, and 3.3 for non-convex case, we can obtain the convergence bounds with appropriate learning rates for CHFL as follows:*

**Strongly convex:** *With following learning rate condition, $\frac{1}{\mu T} \leq \tilde{\eta} \leq \frac{1}{35L}$, we have the convergence rate:*

$$\Pi_{sc} = \tilde{\mathcal{O}} \left( \frac{\sigma^2}{\mu MNRKT} + \frac{L(M^2NR^2 + NR^2 + 1)\sigma_c^2}{M^2N^2K^2\mu^2T^2} + \frac{L\sigma_g^2}{M\mu^2T^2} + \mu D^2 \exp(-\frac{\mu T}{70L}) \right) \quad (4)$$

**General convex:** *With following learning rate condition, $\tilde{\eta} \leq \frac{1}{35L}$. We have the convergence rate:*

$$\Pi_{gc} = \mathcal{O} \left( \frac{\sigma D}{\sqrt{MNRKT}} + \frac{(L(M^2NR^2 + NR^2 + 1)D^4\sigma_c^2)^{\frac{1}{3}}}{(MNRT)^{2/3}} + \frac{(L\sigma_g^2 D^4)^{\frac{1}{3}}}{(MT^2)^{\frac{1}{3}}} + \frac{LD^2}{T} \right) \quad (5)$$

**Non-convex:** *With following learning rate condition, $\tilde{\eta} \leq \frac{1}{35L}$. We have the convergence rate:*

$$\Pi_{nc} = \mathcal{O} \left( \frac{(L\sigma^2 H)^{1/2}}{\sqrt{MNRKT}} + \frac{(L^2 q_\sigma(M, N, R, K)H^2)^{1/3}}{(MNRKT^2)^{1/3}} \left( \sigma_g^2 + \sigma_c^2 \right)^{\frac{1}{3}} + \frac{LH}{T} \right) \quad (6)$$

*where $\mathcal{O}$ omits absolute constants and $\tilde{\mathcal{O}}$ omits absolute constants and polylogarithmic factors. $H := f(\mathbf{x}^0) - f(\mathbf{x}^*)$ for the non-convex case and $D := ||\mathbf{x}^0 - \mathbf{x}^*||$ for the convex case.*

## 4.1 CONVERGENCE RATE

By Corollary 4.2, for a sufficiently large $T$, the convergence rate is determined by the first term induced by SGD variance $\sigma$ for all cases, result in convergence rates of $\tilde{\mathcal{O}}(1/MNRKT)$, $\mathcal{O}(1/\sqrt{MNRKT})$, $\mathcal{O}(1/\sqrt{MNRKT})$ for strongly convex case, general convex case, and non-convex case, respectively. With different gradient descent methods and data heterogeneity scenarios, we have different convergence rates caused by the change of the dominant term. When $\sigma \simeq 0$ with a better SGD variance reduction method De & Goldstein (2016); Alain et al. (2015) or using GD for gradient updates Andrychowicz et al. (2016), the convergence rate can be improved. For example, for the general convex case with $\sigma \simeq 0$, the best convergence rate is $\mathcal{O}(1/T^{2/3})$ when the dominant term depends on $\sigma_c$ and $\sigma_g$ unequally for different decay rates. In the strongly convex case, the best convergence rate can reach to $\tilde{\mathcal{O}}(1/T^2)$ under similar conditions. For the non-convex case, the best convergence rate can achieve $\mathcal{O}(1/T^{2/3})$ depending on $\sigma_c$ and $\sigma_g$ equally for the same decay rate.

## 4.2 EFFECT OF DATA HETEROGENEITY

The inter-edge data heterogeneity significantly affects convergence speed more than intra-edge data heterogeneity in specific conditions for general convex and strongly convex cases, while have equal effects for non-convex case.

• **General convex case.** The decay rate of $\sigma_c$ is $\mathcal{O}(\frac{(M^2NR^2 + NR^2 + 1)^{\frac{1}{3}}}{(MNRT)^{2/3}})$ and the decay rate of $\sigma_g$ is $\mathcal{O}(\frac{1}{(MT^2)^{\frac{1}{3}}})$. When $\frac{(M^2NR^2 + NR^2 + 1)^{\frac{1}{3}}}{(MNRT)^{2/3}} < \frac{1}{(MT^2)^{\frac{1}{3}}}$, i.e., $M < N$, the inter-edge data heterogeneity has more effect on convergence speed than intra-edge data heterogeneity. This finding gives us the insights that **when the number of clients per edge exceeds the total number of edge servers (a common case in practice), reducing inter-edge data heterogeneity $\sigma_g$ can enhance the convergence speed**. For example, if we train a next word prediction model with CHFL for all ages people, and each age has its own word typing habit, it is better to have one edge train the data with all ages to reduce $\sigma_g$ rather than one edge covering one age.

• **Strongly convex case.** We can still achieve a faster convergence speed by reducing $\sigma_g$ when $M < N$ with appropriate settings of $K$ and $R$ to satisfy $MR^2 < NK^2$. This condition ensures that the decay rate of intra-edge data heterogeneity is faster than inter-edge data heterogeneity, i.e. $\frac{M^2NR^2 + NR^2 + 1}{M^2N^2K^2T^2} < \frac{1}{MT^2}$. These results align with the cluster policy where the data distribution among edges tends to be IID, i.e., $\sigma_g \simeq 0$, as suggested by Mhaisen et al. (2021); Deng et al. (2021). However, other works like Liu et al. (2020); Wang et al. (2022) propose a completely opposite cluster policy by grouping clients with similar data to reduce intra-edge data heterogeneity $\sigma_c$ and improve convergence speed. This opposing approach is also effective when $MR^2 > NK^2$, such as when the intra-edge data heterogeneity decays more slowly than the inter-edge heterogeneity.

Based on our convergence analysis, **reducing any form of data heterogeneity, whether intra-edge or inter-edge, can enhance convergence speed. The challenge lies in determining which reduction yields the optimal improvement.** This largely depends on the settings of $M, N, R, K$ and objectives.

### 4.3 Effect of edge round $R$

Based on Corollary 4.2, with non-zero $\sigma$, increasing $R$ speeds up the convergence with sufficient large $T$, but it cannot always benefit convergence speed in other scenarios. For example, as in the strongly convex case, larger edge training round $R$ has a negative effect on the convergence speed when the intra-edge data heterogeneity dominates convergence. Moreover, when the dominant term depends on inter-edge data heterogeneity for both strongly convex and general convex case, increasing $R$ can not affect the convergence speed. For the non-convex case, edge training round $R$ affects both terms of SGD variance $\sigma$ and data heterogeneity, increasing $R$ improves convergence speed.

### 4.4 Participation Pattern

Since partial edge/device participation has more practical interest than full edge/device participation, we also derive the bound for partial participation for strongly convex and general convex cases. Consider only $S \leq M$ edges are randomly selected for training in each global round, in which each edge selects $P \leq N$ clients for local training. Fully participation can achieve a better convergence rate than partial participation. There are additional terms caused by partial participation on both strongly convex and general convex cases, which is consistent with Li & Lyu (2024); Yang et al. (2021). The difference lies in the additional term caused by the edge sampling, i.e., the second term in Eq. 7 and Eq. 8. The inclusion of the two middle terms shows that the number of selected clients and edges can still enhance the convergence speed. Due to the limited space, we take $\phi_{sc}(S, P, K, R, T)$ and $\phi_{gc}(S, P, K, R, T)$ as the last three terms of Eq. 4 and Eq. 5, where $M$ and $N$ are replaced by $S$ and $P$.

$$\Pi_{sc} = \tilde{\mathcal{O}}\left(\frac{\sigma^2}{\mu SPRKT} + \frac{(N-P)\sigma_c^2}{\mu TP(N-1)} + \frac{(M-S)\sigma_g^2}{\mu TS(M-1)} + \phi_{sc}(S, P, K, R, T)\right) \quad (7)$$

$$\Pi_{gc} = \mathcal{O}\left(\frac{\sigma D}{\sqrt{SPRKT}} + \sqrt{\frac{(N-P)D^2\sigma_c^2}{TP(N-1)}} + \sqrt{\frac{(M-S)D^2\sigma_g^2}{TS(M-1)}} + \phi_{gc}(S, P, K, R, T)\right) \quad (8)$$

### 4.5 Discussion and Limitations

As in Li & Lyu (2024); Karimireddy et al. (2020), the cyclic FL experiences client drift resulting from data heterogeneity among clients. In CHFL, we encounter additional edge model drift due to inter-edge heterogeneity. We assume an estimator in each local step, denoted by $\mathbf{g}_{r,k}^{i,j} = \nabla F_j(\mathbf{x}_{r,k}^{i,j}, \xi_{r,k}^{i,j})$. When bounding it, three additional terms due to the edge layer and cyclic pattern make the formula derivation more complex than Li & Lyu (2024). See details in Lemma C.1.1. Our convergence rate achieves the optimal rate compared with other FL and HFL variants. In Tab. 1, FL represents a special case of our CHFL when $N = 1$ and $R = 1$. We maintain the same convergence rate as FL with full participation for larger $T$, across strongly convex Li & Lyu (2024); Karimireddy et al. (2020); Koloskova et al. (2020), general convex and non-convex objectives Li & Lyu (2024). HFL generally outperforms FL even with the same local steps on clients, as HFL benefits from more aggregation Lee et al. (2020). Our convergence rate outperforms other centralized HFLs under standard settings, as all hyperparameters contribute to faster convergence in terms of local steps. In comparison to CFL Cho et al. (2023), we have edge round $R > 1$ to speed up the convergence speed. Furthermore, relative to Liu et al. (2020; 2023; 2022), we include $M$ and $N$ to the dominant term to accelerate convergence speed. For data heterogeneity in Group-FEL Liu et al. (2023), they show that inter-edge data heterogeneity affects convergence speed more than intra-edge data heterogeneity, which is consistent with our findings with specific settings. However, even with the optimal convergence rate with our settings, we only provide the convergence analysis when the edge server performs synchronous FL McMahan et al. (2017), but the asynchronous FL Sprague et al. (2018) in the edge server is more practical in the world for system heterogeneity Li et al. (2021) of clients.

# 5 EXPERIMENTS

## 5.1 EXPERIMENTAL SETTINGS

To validate our theoretical findings, we use the convolutional neural network (CNN) in Krizhevsky et al. (2017) on manually partitioned Non-IID MNIST dataset Wang et al. (2021a), Resnet-32 He et al. (2016) on manually partitioned Non-IID CIFAR-10 dataset Krizhevsky et al. (2009) and Long short-term memory (LSTM) Hochreiter & Schmidhuber (1997) on natural Non-IID dataset Shakespeare McMahan et al. (2017). To impose data heterogeneity in MNIST, we distribute the data evenly into each client in label-based partition following the same process in McMahan et al. (2017). Compared with the vanilla FL, we consider two types of data heterogeneity: intra-edge data heterogeneity $p_c$ and inter-edge data heterogeneity $p_g$. Similar to the $p$ value in Yang et al. (2022; 2021) which represents the number of labels in the edge or client, $p_c$ indicates the number of labels in the client, and $p_g$ describes the number of labels in the edge, respectively. The smaller $p_g$ or $p_c$, the more heterogeneity of the data across edges or clients. We compare five algorithms, SFL Li & Lyu (2024), vanilla FL McMahan et al. (2017), cyclic FL (CFL) Cho et al. (2023), Hierarchical FL (HFL) Liu et al. (2020) and our CHFL with varied data heterogeneity. In particular, CFL is the special case of CHFL with $R = 1$. SFL is the special case of CHFL with N=1 and R=1. We set a total of 10 edge servers and the number of clients is 500. Also, $p_c$ and $p_g$ are 1, 2, 5, and 10, $\eta = 0.01$, batch size $b = 32$, $R = 2$, $K = 2$ and selected edges $P = 2$. To ensure a fair comparison, all algorithms are trained using the same number of local steps on clients instead of communication rounds. Our experiment is conducted with one NVIDIA A100 GPU, 4 CPU cores, and 128 GB memory. The details of models, datasets, and hyper-parameters, and further results of other datasets can be found in Appendix B.

## 5.2 EFFECT OF DATA HETEROGENEITY

We evaluate test accuracy and convergence speed for the MNIST dataset using different algorithms with various data heterogeneity.

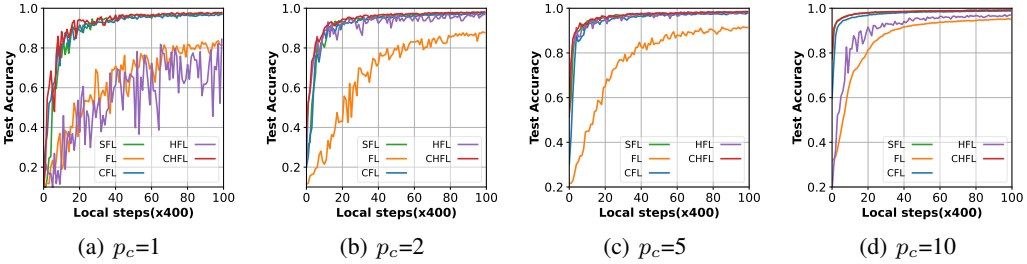

(a) $p_c$=1      (b) $p_c$=2      (c) $p_c$=5      (d) $p_c$=10

Figure 2: Test accuracy w/ Intra-edge Data Heterogeneity on MNIST Dataset

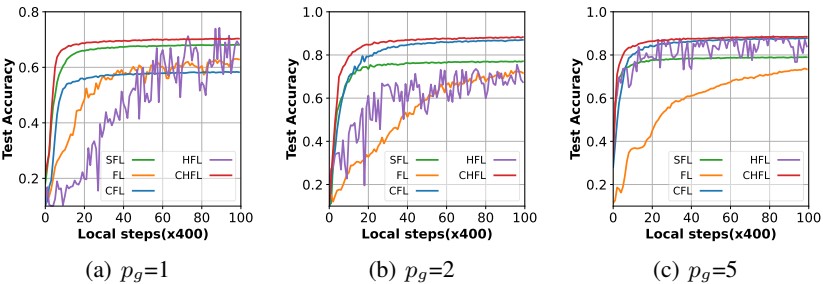

(a) $p_g$=1      (b) $p_g$=2      (c) $p_g$=5

Figure 3: Test Accuracy w/Inter-edge Data Heterogeneity on MNIST Dataset

1) CHFL has comparable or superior convergence speed and accuracy than other algorithms for any of $p_c$ and $p_g$ conditions. Taking $p_c = 1$ for instance, CHFL has better accuracy with 97.84% than SFL with 97.25% that is the highest accuracy among other algorithms. With the variance of inter-edge data heterogeneity in Fig. 3, i.e., $\sigma_c = 0$, CHFL also demonstrates better accuracy than other algorithms. When compared to the second-highest accuracy achieved by other algorithms with varied inter-edge data heterogeneity, CHFL shows an improvement of up to 2% when $p_g = 1$ in Fig.3(a).

2) The impact of inter-edge differences on convergence and accuracy is greater or equal than intra-edge data heterogeneity under our settings. As we can see in Fig.3, we got an accuracy of 70.32% with extreme inter-edge data heterogeneity in Fig.3(a), the accuracy decreased much than the accuracy of 97.25% with extreme intra-edge data heterogeneity in Fig.2(a). The same result we can get when comparing Fig.2(b) and Fig.3(b). We verify the above insight that it is better to reduce inter-edge data heterogeneity, i.e., have each edge share similar data. Compared to other methods, FL and HFL face challenges with extreme data heterogeneity, particularly with inter-edge data heterogeneity. Due to their relatively infrequent aggregation compared to the cyclic pattern, their updates are more prone to bias, leading to greater fluctuations and instability in the model, like Fig. 2(a) and Fig. 3(a). Comparing SFLLi & Lyu (2024) and CFLCho et al. (2023) with cyclic pattern architecture, CHFL has a similar or faster convergence speed compared to these methods, but it achieves faster training in wall-clock time than SFL by enabling client parallel training in edge instead of sequential client training. Compared with CFL, the additional edge layer with edge rounds $R$ helps accelerate the convergence speed.

### 5.3 EFFECT OF EDGE TRAINING ROUND

In Fig. 4 and Fig. 5, we mainly focus on evaluating the impact of $R$ in CHFL. We can see that when we increase edge round $R$ on training, the convergence speed and accuracy can be improved. This result is consistent with our theory result. In Fig. 5, $R$ affects the convergence rate more significantly than in Fig. 4. For example, in Fig.5(a), we have 15% accuracy improvement when we increased $R$ from 1 to 10. The reason is larger $R$ can accelerate the convergence speed and the term of inter-edge data heterogeneity with $p_c = 10$, i.e., $\sigma_c \simeq 0$, has a weaker effect on convergence speed based on our theoretical results.

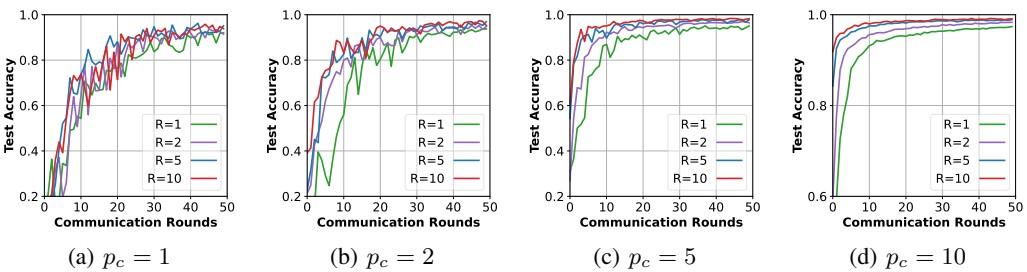

| (a) $p_c = 1$ | (b) $p_c = 2$ | (c) $p_c = 5$ | (d) $p_c = 10$ |

Figure 4: CHFL with $R$ and $p_c$ on MNIST Dataset

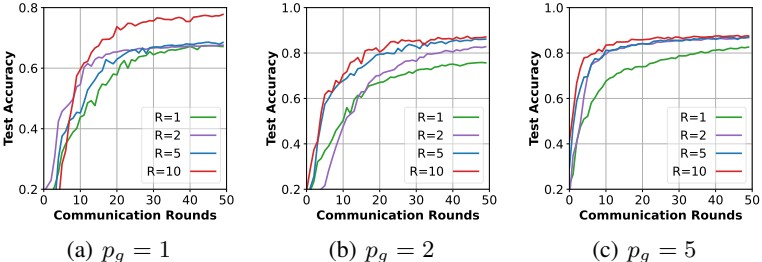

| (a) $p_g = 1$ | (b) $p_g = 2$ | (c) $p_g = 5$ |

Figure 5: CHFL with $R$ and $p_g$ on MNIST Dataset

## 6 CONCLUSION

In this paper, we derive convergence guarantees for CHFL on heterogeneous data for strongly convex, general convex, and non-convex objectives, considering both full and partial participation. Compared to other FL and HFL variants, we verify that our convergence rate is optimal to date. Based on our theoretical results, we find that clustering clients solely by similar or opposing data distributions does not achieve the best improvement in convergence speed. Instead, optimal clustering depends on the system settings under various objectives. We hope the insights in this paper will facilitate the deployment of CHFL in real-world applications.

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

## A    NOTATION TABLE

| Symbol Definitions | |
|---|---|
| Definition | Symbol/Notation |
| $\mathbf{x}$ | model weight |
| C | the total number of clients |
| M | the number of edge servers |
| S | the number of selected edge servers |
| R | edge server training rounds |
| r | the r-th training round in edge server |
| K | client training local steps |
| k | the k-th training local epoch in client |
| N | the number of clients in each edge |
| P | the number of selected clients in each edge |
| $\mathcal{N}_i$ | all clients in edge $i$ |
| $F_j(\mathbf{x})$ | the objective of client $j$ |
| $f_i(\mathbf{x})$ | the objective of edge server $i$ |
| T | global model update rounds |
| t | the t-th global model update round |
| $\sigma^2$ | the SGD variance |
| $\sigma_g^2$ | Inter-edge data heterogeneity |
| $\sigma_c^2$ | Intra-edge data heterogeneity |
| $\eta$ | learning rate in client |

## B    ADDITIONAL EXPERIMENTS

### B.1    DATASETS

We show detailed results of CNN on the MNIST dataset, ResNet32 on the CIFAR10 and LSTM on the Shakespeardataset.Table2 shows the CNN model architecture we use for training.

The settings for model training on the CIFAR10 dataset are followed. Same as the way that we impose the data heterogeneity in the MNIST dataset, we have $p_c$ and $p_g$ to define the intra-edge data heterogeneity and inter-edge data heterogeneity. We having settings for CHFL: edge server number $M = 10$, local client $N = 5$ and selected clients $P = 2$, local steps $K = 2$, learning rate $\eta = 0.01$, batch size $b = 32$. We use ResNet32Orhan (2018) to train local models.

The settings for model training on Shakespear dataset is followed. Since Shakespeare dataset is a natural Non-iid dataset, so we set each role as one client and the total number of clients is 139. Each client have non-i.i.d data with each other, so we can assume it as $p_c = 1, p_g = 1$. We assume the edge server number $M = 18$, there are 8 clients on each edge and the last edge has 3 clients. The learning rate is 0.8 and the batch size is 32. We show the result for four algorithms when selecting $S = 16$ edges in SFL and selecting $P = 1$ clients, select $S = 2$ edges and $P = 8$ clients randomly for other algorithms. The local training step is $K = 2$ and edge training epoch $R = 1, 2$. We use LSTM to train it. Specifically, SFL and CFL set $R = 1$, other two algorithms set $R = 2$. For LSTM architecture, we have an embedding size of 80x8, two LSTM layers with input size 8 and hidden size 256, final linear layer with 256x8.

Table 2: CNN Architecture for MNIST

| Layer Type | Size |
|---|---|
| Convolution+ReLu | 5x5x10 |
| Max Pooling | 2x2 |
| Convolution+ReLu | 5x5x20 |
| Max Pooling | 2x2 |
| Fully Connected+ReLU | 320x50 |
| Fully Connected | 50x10 |

## B.2 ALGORITHMS

Cyclic Federated Learning(CFL) Cho et al. (2023) divides $C$ clients into $M$ non-overlapping client groups. The groups and the order in which they are traversed by the central server are pre-determined and fixed throughout training to simulate a cyclic structure of client participation. In each global round, once one group becomes available, the server would select all or partial clients to train. Once selected, this group can not participate again at least for the next $M - 1$ global rounds. In Fig.6, we have settings for CFL as $C = 6, M = 3, N = 2$.

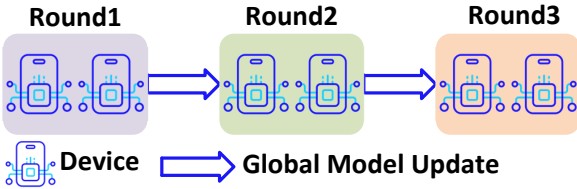

Figure 6: Cyclic Federated Learning

Sequential Federated Learning(SFL) Li & Lyu (2024) sample clients without replacement randomly as the clients' training order. Within a round, each client initializes its model from its previous client and performs $K$ local steps over its local dataset. Then, it passes its own model to the next client until all clients finish their local training. In Fig.7, we have process for SFL in one global round and set total clients number as $C = 6$.

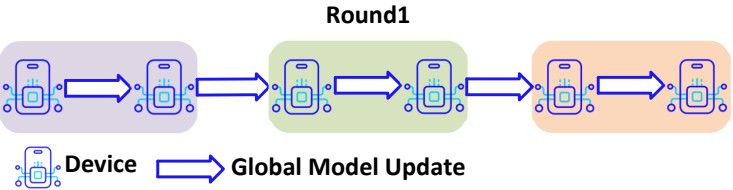

Figure 7: Sequential Federated Learning

To ensure a fair comparison, we establish the following settings for data heterogeneity and training rounds. When comparing these algorithms under different levels of data heterogeneity, we set $p_c$ and $p_g$ values for HFL and CHFL. For CFL, we keep the same edge setting as CHFL but $R = 1$. In contrast, SFL and FL have no inter-edge data heterogeneity, so we maintain the same $p_c$ and $p_g$ values, randomly selecting the same number of clients from the entire clients pool. All architectures are trained using the same number of local steps rather than communication rounds. The detailed results of four algorithms are in Table.3.

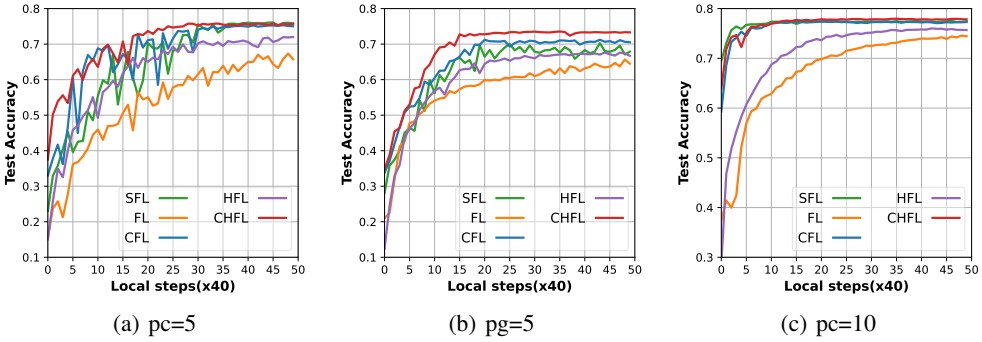

Figure 8: Test Accuracy w/Intra-edge Data Heterogeneity on CIFAR10 Dataset

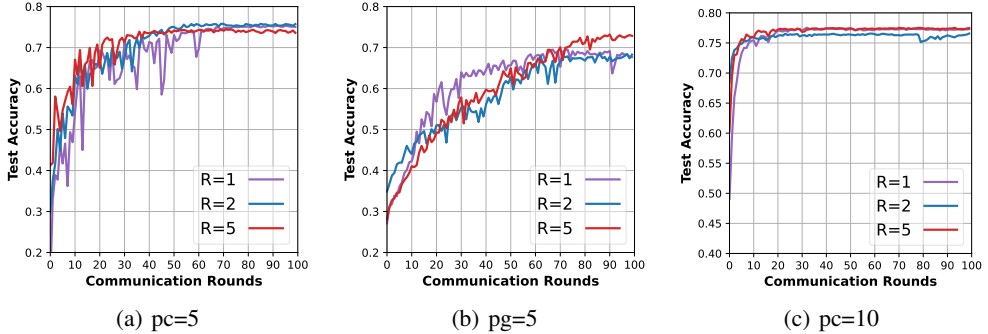

(a) pc=5          (b) pg=5          (c) pc=10

Figure 9: Test Accuracy w/Intra-edge Data Heterogeneity and Edge round on CIFAR10 Dataset

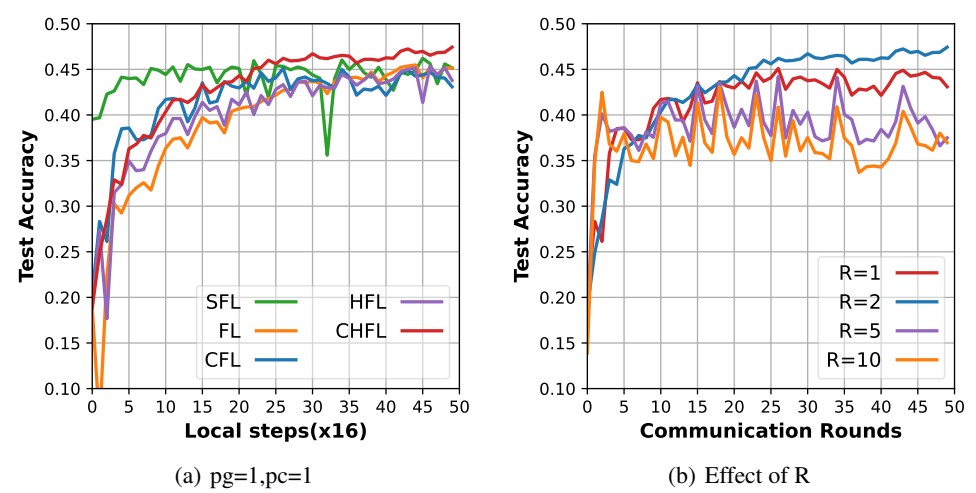

(a) pg=1,pc=1          (b) Effect of R

Figure 10: Test Accuracy w/Inter-edge Data Heterogeneity and Edge round on Shakespeare Dataset

Table 3: Test accuracy for comparison of various algorithms.

| Model/Dataset | Non-I.I.D. index(p) | Algorithms | | | | |
|---|---|---|---|---|---|---|
| | | SFL | FL | CFL | HFL | CHFL |
| CNN/MNIST | $p_c$=1 | 0.975 | 0.8144 | 0.9254 | 0.8234 | **0.9784** |
| | $p_c$=2 | 0.9778 | 0.8773 | 0.9693 | 0.9613 | **0.9812** |
| | $p_c$=5 | 0.9823 | 0.9138 | 0.9784 | **0.9845** | 0.9844 |
| | $p_c$=10 | 0.991 | 0.9509 | 0.9878 | 0.8266 | **0.9928** |
| | $p_g$=1 | 0.6811 | 0.6285 | 0.5824 | 0.5111 | **0.7032** |
| | $p_g$=2 | 0.7708 | 0.5309 | 0.7168 | 0.827 | **0.9699** |
| | $p_g$=5 | 0.8896 | 0.8339 | 0.9775 | 0.9719 | **0.9845** |
| ResNet32/CIFAR10 | $p_c$=1 | 0.1 | 0.1 | 0.1 | 0.1139 | **0.1493** |
| | $p_c$=2 | 0.3425 | 0.5511 | 0.4801 | 0.5857 | **0.5857** |
| | $p_c$=5 | 0.7573 | 0.552 | 0.7503 | 0.72 | **0.758** |
| | $p_c$=10 | 0.7503 | 0.7168 | 0.7729 | 0.7445 | **0.7783** |
| | $p_g$=1 | 0.156 | 0.1 | 0.1 | 0.156 | **0.1518** |
| | $p_g$=2 | 0.3874 | 0.2875 | 0.1967 | 0.2113 | **0.4596** |
| | $p_g$=5 | 0.6525 | 0.5765 | 0.7059 | 0.6579 | **0.7334** |
| LSTM/Shakespeare | $p_c = 1, p_g = 1$ | 0.4516 | 0.4516 | 0.4308 | 0.4379 | **0.4743** |

# C PROOFS

## C.1 PROOFS OF THEOREM 4.1

In this section, we give the proofs in detail for full and partial edge/client participation with HFL with cyclic pattern for three cases: strongly convex, general convex and non-convex. We will show them respectively. Our proofs are based on some identities and inequalities from C.2 and C.3 in Li & Lyu (2024), please check it for reference.

### C.1.1 STRONGLY CONVEX

**Lemma C.1.** *Let Assumptions 3.1, 3.2,3.4 hold and assume that all the local objectives are $\mu$-strongly convex. If the learning rate satisfies $\eta \leq \frac{1}{35LSPRTK}$, then it holds that*

$$\mathbb{E}\left[\left\|\mathbf{x}^{(t+1)} - \mathbf{x}^*\right\|^2\right] \leq \left(1 - \frac{\mu SPKR\eta}{2}\right)\left\|\mathbf{x}^t - \mathbf{x}^*\right\|^2 + 7SPRK\eta^2\sigma^2 + 7S^2P^2R^2K^2\eta^2\frac{(N-P)}{P(N-1)}\sigma_c^2$$

$$+ \frac{21}{S(M-1)}\eta^2\sigma_g^2 - \frac{8}{5}SPKR\eta D_F\left(\mathbf{x}^t, \mathbf{x}^*\right) + 42LS^2P^2R^2K^2D_f\left(\mathbf{x}^t, \mathbf{x}^*\right)$$

$$+ \left(2L\eta + 7L^2SPRK\eta^2\right)\sum_{i,j,r,k}\mathbb{E}\left[\left\|\mathbf{x}_{r,k,t}^{i,j} - \mathbf{x}^t\right\|^2\right]$$

*where $D_F\left(\mathbf{x}, \mathbf{x}^*\right) = \mathbb{E}\left[\left\|\nabla F_j(\mathbf{x}) - \nabla F_j\left(\mathbf{x}^*\right)\right\|^2\right]$ and $D_f\left(\mathbf{x}, \mathbf{x}^*\right) = \mathbb{E}\left[\left\|\nabla f_i(\mathbf{x}) - \nabla f_i\left(\mathbf{x}^*\right)\right\|^2\right]$.*

*Proof.* At $t$ global round, the global model update of CHFL after one complete training round:

$$\Delta\mathbf{x} = \mathbf{x}^{t+1} - \mathbf{x}^t = -\eta\sum_{i=0}^{S-1}\sum_{j=0}^{P-1}\sum_{r=0}^{R-1}\sum_{k=0}^{K-1}\mathbf{g}_{r,k}^{i,j}$$

where $\mathbf{g}_{r,k}^{i,j} = \nabla F_j(\mathbf{x}_{r,k}^{i,j}, \xi_{r,k}^{i,j})$, thus:

$$\mathbb{E}[\Delta\mathbf{x}] = -\eta\sum_{i,j,r,k}\mathbb{E}\left[\nabla F_j\left(\mathbf{x}_{r,k}^{i,j}\right)\right]$$

We focus on a single training round and drop the superscripts $t$:

$$\mathbb{E}\left[\left\|\mathbf{x} + \Delta\mathbf{x} - \mathbf{x}^*\right\|^2\right]$$

$$= \left\|\mathbf{x} - \mathbf{x}^*\right\|^2 + 2\mathbb{E}\left[\langle\mathbf{x} - \mathbf{x}^*, \Delta\mathbf{x}\rangle\right] + \mathbb{E}\left[\left\|\Delta\mathbf{x}\right\|^2\right]$$

$$= \left\|\mathbf{x} - \mathbf{x}^*\right\|^2 \underbrace{- 2\eta\sum_{i,j,r,k}\mathbb{E}\left[\left\langle\nabla F_j\left(\mathbf{x}_{r,k}^{i,j}\right), \mathbf{x} - \mathbf{x}^*\right\rangle\right]}_{A} + \eta^2\underbrace{\mathbb{E}\left[\left\|\sum_{i,j,r,k}g_{r,k}^{i,j}\right\|^2\right]}_{B}$$

Using Lemma 2 inLi & Lyu (2024) to bound $A$:

$$-2\eta\sum_{i,j,r,k}\mathbb{E}\left[\left\langle\nabla F_j\left(\mathbf{x}_{r,k}^{i,j}\right), \mathbf{x} - \mathbf{x}^*\right\rangle\right]$$

$$\leq -2\eta\sum_{i,j,r,k}\mathbb{E}\left[F_j(\mathbf{x}) - F_j\left(\mathbf{x}^*\right) + \frac{\mu}{4}\left\|\mathbf{x}^* - \mathbf{x}\right\|^2 - L\left\|\mathbf{x} - \mathbf{x}_{r,k}^{i,j}\right\|^2\right]$$

$$\leq -2SPKR\eta D_F\left(\mathbf{x}, \mathbf{x}^*\right) - \frac{1}{2}\mu SPKR\eta\|\mathbf{x} - \mathbf{x}^*\|^2 + 2L\eta\sum_{i,j,r,k}\left\|\mathbf{x} - \mathbf{x}_{r,k}^{i,j}\right\|^2$$

Bounding $B$ using Jensen's inequality in C.2 Li & Lyu (2024), We observe that the three additional terms ⑤⑥⑦ are generated by the edge layer and the cyclic pattern.:

$$\mathbb{E}\left[\left\|\sum_{i,j,r,k} g_{r,k}^{i,j}\right\|^2\right]$$

$$= \mathbb{E}\left[\left\|\sum_{i,j,r,k}\left\{g_{r,k}^{i,j} - \nabla F_j\left(\mathbf{x}_{r,k}^{i,j}\right) + \nabla F_j\left(\mathbf{x}_{r,k}^{i,j}\right) - \nabla F_j(\mathbf{x})\right.\right.\right.$$

$$\left.\left.\left. + \nabla F_j(\mathbf{x}) - \nabla F_j\left(\mathbf{x}^*\right) + \nabla F_j\left(\mathbf{x}^*\right) + \nabla f_i(\mathbf{x}) - \nabla f_i\left(\mathbf{x}^*\right) + \nabla f_i\left(\mathbf{x}^*\right) - \nabla f_i(\mathbf{x})\right\}\right\|^2\right]$$

$$\leq \underbrace{7\mathbb{E}\left[\left\|\sum_{i,j,r,k}\left(g_{r,k}^{i,j} - \nabla F_j\left(\mathbf{x}_{r,k}^{i,j}\right)\right)\right\|^2\right]}_{①} + \underbrace{7\mathbb{E}\left[\left\|\sum_{i,j,r,k}\left(\nabla F_j\left(\mathbf{x}_{r,k}^{i,j}\right) - \nabla F_j(\mathbf{x})\right)\right\|^2\right]}_{②}$$

$$+ \underbrace{7\mathbb{E}\left[\left\|\sum_{i,j,r,k}\left[\nabla F_j\left(\mathbf{x}\right) - \nabla F_j\left(\mathbf{x}^*\right)\right]\right\|^2\right]}_{③} + \underbrace{7\mathbb{E}\left[\left\|\sum_{i,j,r,k}\nabla F_j\left(\mathbf{x}^*\right)\right\|^2\right]}_{④}$$

$$+ \underbrace{7\mathbb{E}\left[\left\|\sum_{i,j,r,k}\left[\nabla f_i\left(\mathbf{x}\right) - \nabla f_i\left(\mathbf{x}^*\right)\right]\right\|^2\right]}_{⑤} + \underbrace{7\mathbb{E}\left[\left\|\sum_{i,j,r,k}\nabla f_i\left(\mathbf{x}^*\right)\right\|^2\right]}_{⑥}$$

$$+ \underbrace{7\mathbb{E}\left[\left\|\sum_{i,j,r,k}\left(-\nabla f_i\left(\mathbf{x}\right)\right)\right\|^2\right]}_{⑦}$$

Bounding ①:

$$7\mathbb{E}\left[\left\|\sum_{i,j,r,k}\left(g_{r,k}^{i,j} - \nabla F_j\left(\mathbf{x}_{r,k}^{i,j}\right)\right)\right\|^2\right] \leq 7SPRK\sigma^2$$

Bounding ②:

$$7\mathbb{E}\left[\left\|\sum_{i,j,r,k}\left(\nabla F_j\left(\mathbf{x}_{r,k}^{i,j}\right) - \nabla F_j(\mathbf{x})\right)\right\|^2\right] \leq 7L^2SPRK\sum_{i,j,r,k}\mathbb{E}\left[\left\|\mathbf{x}_{r,k}^{i,j} - \mathbf{x}\right\|^2\right]$$

Bounding ③:

$$7\mathbb{E}\left[\left\|\sum_{i,j,r,k}\left[\nabla F_j\left(\mathbf{x}\right) - \nabla F_j\left(\mathbf{x}^*\right)\right]\right\|^2\right] \leq 14LSPRK\sum_{i,j,r,k}\mathbb{E}\left[D_{F_j}\left(\mathbf{x},\mathbf{x}^*\right)\right] \quad \leq 14LS^2P^2R^2K^2D_F\left(\mathbf{x},\mathbf{x}^*\right)$$

Bounding ④:

$$7\mathbb{E}\left[\left\|\sum_{i,j,r,k}\nabla F_j\left(\mathbf{x}^*\right)\right\|^2\right] \leq 7S^2R^2P^2K^2 \times \frac{N-P}{P(N-1)}\sigma_c^2$$

Bounding ⑤:

$$7\mathbb{E}\left[\left\|\sum_{i,j,r,k}\left[\nabla f_i\left(\mathbf{x}\right) - \nabla f_i\left(\mathbf{x}^*\right)\right]\right\|^2\right] \leq 14LSPRK\sum_{i,j,r,k}\mathbb{E}\left[D_{fi}\left(\mathbf{x},\mathbf{x}^*\right)\right] \leq 14LS^2P^2R^2K^2D_f\left(\mathbf{x},\mathbf{x}^*\right)$$

Bounding ⑥:

$$7\mathbb{E}\left[\left\|\sum_{i,j,r,k}\nabla f_i\left(\mathbf{x}^*\right)\right\|^2\right] \leq 7P^2K^2R^2\mathbb{E}\left[\left\|\sum_{i=0}^{S-1}\nabla f_i\left(\mathbf{x}^*\right)\right\|^2\right] \leq 7P^2K^2R^2\frac{(M-S)}{S(M-1)}\sigma_g^2$$

Bounding ⑦:

$$7\mathbb{E}\left[\left\|\sum_{i,j,r,k}\left(-\nabla f_i\left(\mathbf{x}\right)\right)\right\|^2\right] = 7\mathbb{E}\left[\left\|\sum_{i,j,r,k}\left(-\nabla f_i\left(\mathbf{x}\right) + \nabla f_i\left(\mathbf{x}^*\right) - \nabla f_i\left(\mathbf{x}^*\right)\right)\right\|^2\right]$$

$$\leq 7\mathbb{E}\left[\left\|\sum_{i,j,r,k}\left(\nabla f_i\left(\mathbf{x}^*\right) - \nabla f_i\left(\mathbf{x}\right) - \nabla f_i\left(\mathbf{x}^*\right)\right)\right\|^2\right]$$

$$\leq \underbrace{14\mathbb{E}\left[\left\|\sum_{i,j,r,k}\left(\nabla f_i\left(\mathbf{x}^*\right) - \nabla f_i\left(\mathbf{x}\right)\right)\right\|^2\right]}_{\text{Based on ⑤}} + \underbrace{14\mathbb{E}\left[\left\|\sum_{i,j,r,k}\nabla f_i\left(\mathbf{x}^*\right)\right\|^2\right]}_{\text{Based on ⑥}}$$

$$\leq 28LS^2P^2R^2K^2D_f\left(\mathbf{x},\mathbf{x}^*\right) + \frac{14P^2K^2R^2(M-S)}{S(M-1)}\sigma_g^2$$

Then substituting above seven bounds into $B$:

$$\mathbb{E}\left[\left\|\sum_{i,j,r,k}g_{r,k}^{i,j}\right\|^2\right] \leq 7SPRK\sigma^2 + 7L^2SPRK\sum_{i,j,r,k}\mathbb{E}\left[\left\|\mathbf{x}_{r,k}^{i,j} - \mathbf{x}\right\|^2\right] + 14LS^2P^2R^2K^2D_F\left(\mathbf{x},\mathbf{x}^*\right)$$

$$+ 7S^2R^2 \times P^2K^2 \times \frac{N-P}{P(N-1)}\sigma_c^2 + 14LS^2P^2R^2K^2D_f\left(\mathbf{x},\mathbf{x}^*\right)$$

$$+ 7P^2K^2R^2\frac{(M-S)}{S(M-1)}\sigma_g^2 + 28LS^2P^2R^2K^2D_f\left(\mathbf{x},\mathbf{x}^*\right) + \frac{14P^2K^2R^2(M-S)^2}{S(M-1)}\sigma_g^2$$

$$= 7SPRK\sigma^2 + 7L^2SPRK\sum_{i,j,r,k}\mathbb{E}\left[\left\|\mathbf{x}_{r,k}^{i,j} - \mathbf{x}\right\|^2\right] + 14LS^2P^2R^2K^2D_F\left(\mathbf{x},\mathbf{x}^*\right)$$

$$+ 7S^2R^2P^2K^2 \times \frac{N-P}{P(N-1)}\sigma_c^2 + 42LS^2P^2R^2K^2D_f\left(\mathbf{x},\mathbf{x}^*\right)$$

$$+ \frac{21P^2K^2R^2S^2(M-S)}{S(M-1)}\sigma_g^2$$

Substituting $A$ and $B$ into following equation:

$$\mathbb{E}\left[\|\mathbf{x} + \Delta\mathbf{x} - \mathbf{x}^*\|^2\right]$$

$$\leq \|\mathbf{x} - \mathbf{x}^*\|^2 - 2\eta \sum_{i,j,r,k} \mathbb{E}\left[\left\langle \nabla F_j\left(\mathbf{x}_{r,k}^{i,j}\right), \mathbf{x} - \mathbf{x}^*\right\rangle\right] + \eta^2 \mathbb{E}\left[\left\|\sum_{i,j,r,k} g_{r,k}^{i,j}\right\|^2\right]$$

$$\leq \|\mathbf{x} - \mathbf{x}^*\|^2 - 2SPKR\eta D_F\left(\mathbf{x}, \mathbf{x}^*\right) - \frac{1}{2}\mu SPKR\eta\|\mathbf{x} - \mathbf{x}^*\|^2 + 2L\eta \sum_{i,j,r,k} \mathbb{E}\left[\left\|\mathbf{x} - \mathbf{x}_{r,k}^{i,j}\right\|^2\right]$$

$$+ 7SPRK\eta^2\sigma^2 + 7L^2SPRK\eta^2 \sum_{i,j,r,k} \mathbb{E}\left[\left\|\mathbf{x}_{r,k}^{i,j} - \mathbf{x}\right\|^2\right] + 14LS^2P^2R^2K^2D_F\left(\mathbf{x}, \mathbf{x}^*\right)$$

$$+ 7S^2R^2 \times P^2K^2\eta^2 \times \frac{N-P}{P(N-1)}\sigma_c^2 + 42LS^2P^2R^2K^2\eta^2 D_f\left(\mathbf{x}, \mathbf{x}^*\right) + \frac{21P^2K^2R^2S^2(M-S)}{S(M-1)}\eta^2\sigma_g^2$$

$$\leq \left(1 - \frac{\mu MPKR\eta}{2}\right)\|\mathbf{x} - \mathbf{x}^*\|^2 + 7SPRK\eta^2\sigma^2 + 7S^2P^2R^2K^2\eta^2\frac{(N-P)}{P(N-1)}\sigma_c^2$$

$$+ \frac{21}{S(M-1)}\eta^2\sigma_g^2 - \frac{8}{5}SPKR\eta D_F\left(\mathbf{x}, \mathbf{x}^*\right) + 42LS^2P^2R^2K^2D_f\left(\mathbf{x}, \mathbf{x}^*\right)$$

$$+ \left(2L\eta + 7L^2SPRK\eta^2\right)\underbrace{\sum_{i,j,r,k} \mathbb{E}\left[\left\|\mathbf{x}_{r,k}^{i,j} - \mathbf{x}\right\|^2\right]}_{\text{client drift}}$$

**Lemma C.2.** *Let Assumptions 3.1, 3.2,3.4 hold and assume that all the local objectives are $\mu$-strongly convex. If the learning rate satisfies $\eta \leq \frac{1}{35LSPRTK}$, then the client shift can be bounded as:*

$$\mathbb{E}_t \leq \frac{71}{10}\eta^2 q_B\sigma^2 + \frac{71}{5}Lq_{B^2}\eta^2 D_F\left(\mathbf{x}, \mathbf{x}^*\right) + \frac{71}{10}q_c\eta^2\sigma_c^2 + 22q_g\eta^2\sigma_g^2 + 43Lq_{B^2}\eta^2 D_f\left(\mathbf{x}, \mathbf{x}^*\right) \quad (9)$$

Follow Lemma 6 in Li & Lyu (2024) to bound client drift:

$$\mathbb{E}_t = \sum_{i,j,r,k} \mathbb{E}\left[\left\|\mathbf{x}_{r,k}^{i,j} - \mathbf{x}\right\|^2\right] \quad (10)$$

where $r'(i) = \begin{cases} R & i' < i-1 \\ r-1 & i' = i-1 \end{cases}$, $j'(i) = \begin{cases} P & i' < i-1 \\ j & \text{the } j\text{th client.} \end{cases}$, $k'(i) = \begin{cases} K-1 & i' < i-1 \\ k-1 & i' = i-1 \end{cases}$.

$$\mathbb{E}\left[\left\|\mathbf{x}_{r,k}^{i,j} - \mathbf{x}\right\|^2\right] \leq \mathbb{E}\left[\left\|-\eta \sum_{i',r',j',k'} g_{r',k'}^{i',j'}\right\|^2\right]$$

$$\leq \eta^2 \mathbb{E}\left[\left\|\sum_{i',j',r',k'}\left(g_{r',k'}^{i',j'} - \nabla F_j\left(\mathbf{x}_{r',k'}^{i',j'}\right) + \nabla F_j\left(\mathbf{x}_{r',k'}^{i',j'}\right) - \nabla F_j\left(\mathbf{x}\right) + \nabla F_j\left(\mathbf{x}\right) - \nabla F_j\left(\mathbf{x}^*\right)\right.\right.\right.$$

$$\left.\left.\left. + \nabla F_j\left(\mathbf{x}^*\right) + \nabla f_i\left(\mathbf{x}\right) - \nabla f_i\left(\mathbf{x}^*\right) + \nabla f_i\left(\mathbf{x}^*\right) - \nabla f_i\left(\mathbf{x}\right)\right)\right\|^2\right]$$

$$\leq 7\eta^2 \mathbb{E}\left[\left\|\sum_{i',j',r',k'}\left(g_{r',k'}^{i',j'} - \nabla F_j\left(\mathbf{x}_{r',k'}^{i',j'}\right)\right)\right\|^2\right] + 7\eta^2 \mathbb{E}\left[\left\|\sum_{i',j',r',k'}\left(\nabla F_j\left(\mathbf{x}_{r',k'}^{i',j'}\right) - \nabla F_j(\mathbf{x})\right)\right\|^2\right]$$

$$+ 7\eta^2 \mathbb{E}\left[\left\|\sum_{i',j',r',k'} [\nabla F_j(\mathbf{x}) - \nabla F_j(\mathbf{x}^*)]\right\|^2\right] + 7\eta^2 \mathbb{E}\left[\left\|\sum_{i',j',r',k'} \nabla F_j(\mathbf{x}^*)\right\|^2\right]$$

$$+ 7\eta^2 \mathbb{E}\left[\left\|\sum_{i',j',r',k'} [\nabla f_i(\mathbf{x}) - \nabla f_i(\mathbf{x}^*)]\right\|^2\right] + 7\eta^2 \mathbb{E}\left[\left\|\sum_{i',j',r',k'} \nabla f_i(\mathbf{x}^*)\right\|^2\right]$$

$$+ 7\eta^2 \mathbb{E}\left[\left\|\sum_{i',j',r',k'} (-\nabla f_i(\mathbf{x}))\right\|^2\right]$$

$$\leq 7\eta^2 \sum_{i',j',r',k'} \mathbb{E}\left[\left\|g_{r',k'}^{i',j'} - \nabla F_j\left(\mathbf{x}_{r',k'}^{i',j'}\right)\right\|^2\right] + 7\eta^2 B_{i,j,r,k} \sum_{i',j',r',k'} \mathbb{E}\left[\left\|\nabla F_j\left(\mathbf{x}_{r',k'}^{i',j'}\right) - \nabla F_j(\mathbf{x})\right\|^2\right]$$

$$+ 7\eta^2 B_{i,j,r,k} \sum_{i',j',r',k'} \mathbb{E}\left[\|\nabla F_j(\mathbf{x}) - \nabla F_j(\mathbf{x}^*)\|^2\right] + 7\eta^2 \mathbb{E}\left[\left\|\sum_{i',j',r',k'} \nabla F_j(\mathbf{x}^*)\right\|^2\right]$$

$$+ 7\eta^2 B_{i,j,r,k} \sum_{i',j',r',k'} \mathbb{E}\left[\|\nabla f_i(\mathbf{x}) - \nabla f_i(\mathbf{x}^*)\|^2\right] + 7\eta^2 \mathbb{E}\left[\left\|\sum_{i',j',r',k'} \nabla f_i(\mathbf{x}^*)\right\|^2\right]$$

$$+ 14\eta^2 B_{i,j,r,k} \sum_{i',j',r',k'} \mathbb{E}\left[\|\nabla f_i(\mathbf{x}) - \nabla f_i(\mathbf{x}^*)\|^2\right] + 14\eta^2 \mathbb{E}\left[\left\|\sum_{i',j',r',k'} \nabla f_i(\mathbf{x}^*)\right\|^2\right]$$

$$\leq 7B_{i,j,r,k}\eta^2\sigma^2 + 7L^2\eta^2 B_{i,j,r,k} \sum_{i',j',r',k'} \mathbb{E}\left[\left\|\mathbf{x}_{r',k'}^{i',j'} - \mathbf{x}\right\|^2\right] + 14L\eta^2 B_{i,j,r,k}^2 D_F(\mathbf{x}, \mathbf{x}^*)$$

$$+ 7\eta^2 \mathbb{E}\left[\left\|\sum_{i',j',r',k'} \nabla F_j(\mathbf{x}^*)\right\|^2\right] + 42L\eta^2 B_{i,j,r,k}^2 D_f(\mathbf{x}, \mathbf{x}^*) + 21\eta^2 \mathbb{E}\left[\left\|\sum_{i',j',r',k'} \nabla f_i(\mathbf{x}^*)\right\|^2\right]$$

where $\sum_{i',j',r',k'} 1 = B_{i,j,r,k} = (i-1)RPK + (r-1)PK + k - 1$ and $\sum_{i',j',r',k'}\sum_{i',j',r',k'} 1 = B_{i,j,r,k}^2$.

Then, returning to $\mathbb{E}_t = \sum_{i,j,r,k} \mathbb{E}\left[\left\|\mathbf{x}_{r,k}^{i,j} - \mathbf{x}\right\|^2\right]$, we have

$$\mathbb{E}_t \leq 7\eta^2\sigma^2 \sum_{i,j,r,k} B_{i,j,r,k} + 7L^2\eta^2 \sum_{i,j,r,k} B_{i,j,r,k} \sum_{i',j',r',k'} \mathbb{E}\left[\left\|\mathbf{x}_{r',k'}^{i',j'} - \mathbf{x}\right\|^2\right]$$

$$+ 14L\eta^2 \sum_{i,j,r,k} B_{i,j,r,k}^2 D_F(\mathbf{x}, \mathbf{x}^*) + 7\eta^2 \underbrace{\sum_{i,j,r,k} \mathbb{E}\left[\left\|\sum_{i',j',r',k'} \nabla F_j(\mathbf{x}^*)\right\|^2\right]}_{\text{⑧}}$$

$$+ 42L\eta^2 \sum_{i,j,r,k} B_{i,j,r,k}^2 D_f(\mathbf{x}, \mathbf{x}^*) + 21\eta^2 \underbrace{\sum_{i,j,r,k} \mathbb{E}\left[\left\|\sum_{i',j',r',k'} \nabla f_i(\mathbf{x}^*)\right\|^2\right]}_{\text{⑨}}$$

As in Li & Lyu (2024); Karimireddy et al. (2020), the cyclic FL experiences client drift resulting from data heterogeneity among clients. If we only have client drift induced by cyclic pattern in FL, we only have one term ⑧ to be bounded, and there are only $K$ and $N$ that affect the gradient updates. In CHFL, we encounter additional edge model drift due to inter-edge heterogeneity. Then, we have

an extra term ⑨ to be bounded. Moreover, for both ⑧ and ⑨, we have additional $R$ and $M$, making the bounding formula more complex. Since we assume sampling without replacement for edges and clients, we follow Lemma 4 Li & Lyu (2024) with $\bar{\mathbf{x}} = \nabla F(\mathbf{x}^*) = 0$ for⑧ and $\bar{\mathbf{x}} = \nabla f(\mathbf{x}^*) = 0$ for ⑨ to bound them. .

$$\mathbb{E}\left[\left\|\sum_{i',j',r',k'} \nabla F_j\left(\mathbf{x}^*\right) - \nabla F\left(\mathbf{x}^*\right)\right\|^2\right] \tag{11}$$

$$= \mathbb{E}\left[\left\|\sum_{j=0}^{N-1} KR\left(i-1\right)\left[\nabla F_j\left(\mathbf{x}^*\right) - \nabla F\left(\mathbf{x}^*\right)\right]\right.\right. \tag{12}$$

$$+ K\left(r-1\right)\sum_{j=0}^{N-1}\left[\nabla F_j\left(\mathbf{x}^*\right) - \nabla F\left(\mathbf{x}^*\right)\right] + k\left[\nabla F_j\left(\mathbf{x}^*\right) - \nabla F\left(\mathbf{x}^*\right)\right]\left.\left.\right\|^2\right] \tag{13}$$

$$= K^2 R^2(i-1)^2 \mathbb{E}\left\|\sum_{j=0}^{N-1}\left[\nabla F_j\left(\mathbf{x}^*\right) - \nabla F\left(\mathbf{x}^*\right)\right]\right\|^2 + K^2(r-1)^2\mathbb{E}\left[\left\|\sum_{j=0}^{N-1}\left(\nabla F_j\left(\mathbf{x}^*\right) - \nabla F\left(\mathbf{x}^*\right)\right)\right\|^2\right] \tag{14}$$

$$+ k^2\mathbb{E}\left[\left\|\nabla F_j\left(\mathbf{x}^*\right) - \nabla F\left(\mathbf{x}^*\right)\right\|^2\right] \tag{15}$$

$$+ 2KR\left(i-1\right)K\left(r-1\right)\left[\left\langle\sum_{j=0}^{N-1}\left(\nabla F_j\left(\mathbf{x}^*\right) - \nabla F\left(\mathbf{x}^*\right)\right), \sum_{j=0}^{N-1}\left[\nabla F_j\left(\mathbf{x}^*\right) - \nabla F\left(\mathbf{x}^*\right)\right]\right\rangle\right] \tag{16}$$

$$+ 2KR\left(i-1\right)k\mathbb{E}\left[\left\langle\sum_{j=0}^{N-1}\left(\nabla F_j\left(\mathbf{x}^*\right) - \nabla F\left(\mathbf{x}^*\right)\right), \left(\nabla F_j\left(\mathbf{x}^*\right) - \nabla F\left(\mathbf{x}^*\right)\right)\right\rangle\right] \tag{17}$$

$$+ K\left(r-1\right)k\mathbb{E}\left[\left\langle\sum_{j=0}^{N-1}\left(\nabla F_j\left(\mathbf{x}^*\right) - \nabla F\left(\mathbf{x}^*\right)\right), \nabla F_j\left(\mathbf{x}^*\right) - \nabla F\left(\mathbf{x}^*\right)\right\rangle\right] \tag{18}$$

$$\leq K^2 R^2(i-1)^2 \times \frac{P^2\left(N-P\right)}{P\left(N-1\right)}\sigma_c^2 + K^2(r-1)^2 P^2 \times \frac{N-P}{P\left(N-1\right)}\sigma_c^2 + k^2\sigma_c^2 \tag{19}$$

$$+ KR\left(i-1\right)K\left(r-1\right)P\frac{\sigma_c^2}{N-1} + 2KR\left(i-1\right)kP \times \frac{\sigma_c^2}{N-1} + K\left(r-1\right)kP \times \frac{\sigma_c^2}{N-1} \tag{20}$$

$$= \left\{\frac{K^2 R^2(i-1)^2 P\left(N-P\right)}{N-1} + \frac{K^2(r-1)^2\left(N-P\right)P}{N-1} + k^2 - \frac{2K^2 R\left(i-1\right)\left(r-1\right)P}{N-1}\right.$$

$$\left. - \frac{2KR\left(i-1\right)kP}{N-1} - \frac{K\left(r-1\right)kP}{N-1}\right\}\sigma_c^2$$

Then we can bound ⑧:

$$\sum_{i,j,r,k}\mathbb{E}\left[\left\|\sum_{i',j',r',k'}\nabla F_j\left(\mathbf{x}^*\right)\right\|^2\right]$$

$$\leq \frac{K^3 R^3 P^2 \times 2S^3}{6} + \frac{K^3 P^2 S \times 2R^3}{6} + \frac{SPR \times 2K^3}{6} \leq \frac{1}{3}SPRK^3\underbrace{\left(S^2 PR^2 + PR^2 + 1\right)}_{q_c(S,P,R,K)}\sigma_c^2$$

Still follow Lemma 4 in Li & Lyu (2024), we give the bound for gradients of edge server ⑨,

$$\sum_{i,j,r,k} \mathbb{E}\left[\left\|\sum_{i',j',r',k'} \nabla f_i\left(\mathbf{x}^*\right)\right\|^2\right]$$

$$\leq \frac{P(P-1)(2P-1)}{6} \times \frac{K(K-1)(2K-1)}{6} \times \sum_i \sum_r \mathbb{E}\left[\left\|\sum_{i',r'}\left(\nabla f_i(\mathbf{x}) - \nabla f(\mathbf{x})\right)\right\|^2\right]$$

$$\leq \frac{PK(P-1)(K-1)(2P-1)(2K-1)}{36} \times \frac{1}{2} S^2 R^3 \sigma_g^2 \leq \underbrace{\frac{1}{18} P^3 K^3 R^3 S^2}_{q_g(S,P,R,K)} \sigma_g^2$$

To bound $\mathbb{E}_t$, we first bound following terms:

$$\sum_{i,j,r,k} B_{i,j,rk} = \sum_{i,j,r,k} \{(i-1)RPK + (r-1)PK + k-1\}$$

$$\leq RPK \times \frac{S(S-1)}{2} \times RPK + PK \times SPK \times \frac{R(R-1)}{2} + SPR + \frac{K(K-1)}{2}$$

$$\leq \underbrace{\frac{S(S-1)R^2P^2K^2}{2} + \frac{R(R-1)}{2}SP^2K^2 + \frac{K(K-1)}{2}SPR}_{q_B(S,P,R,K)} \qquad (21)$$

$$\sum_{i,j,r,k} B_{i,j,r,k}^2 \leq \underbrace{\begin{matrix} \frac{R^3P^3K^3S^2(S-1)}{3} + SP^3K^3 \times \frac{R^2(R-1)}{3} + \frac{K^2(K-1)SPR}{3} + \frac{R^2(R-1)S(S-1)P^3K^3}{2} \\ + \frac{R^2P^2K^2S(S-1)(K-1)}{2} + \frac{R(R-1)K^2(K-1)P^2S}{2} \end{matrix}}_{q_{B^2}(S,P,R,K)}$$

$$(22)$$

We have following $\mathbb{E}_t$,

$$\mathbb{E}_t \leq 7\eta^2\sigma^2 \sum_{i,j,r,k} B_{i,j,r,k} + 7L^2\eta^2 \sum_{i,j,r,k} B_{i,j,r,k} \sum_{i',j',r',k'} \mathbb{E}\left[\left\|\mathbf{x}_{r',k'}^{i',j'} - \mathbf{x}\right\|^2\right]$$

$$+ 14L\eta^2 \sum_{i,j,r,k} B_{i',j',r',k'}^2 D_F\left(\mathbf{x},\mathbf{x}^*\right) + 7\eta^2 \sum_{i,j,r,k} \mathbb{E}\left[\left\|\sum_{i',j',r',k'} \nabla F_j\left(\mathbf{x}^*\right)\right\|^2\right]$$

$$+ 42L\eta^2 \sum_{i,j,r,k} B_{i,j,r,k}^2 D_f\left(\mathbf{x},\mathbf{x}^*\right) + 21\eta^2 \sum_{i,j,r,k} \mathbb{E}\left[\left\|\sum_{i',j',r',k'} \nabla f_i\left(\mathbf{x}^*\right)\right\|^2\right]$$

$$\leq 7\eta^2\sigma^2 q_B + 7L^2\eta^2 \sum_{i,j,r,k} q_B \mathbb{E}_t + 14L\eta^2 q_{B^2} D_F\left(\mathbf{x},\mathbf{x}^*\right)$$

$$+ 7\eta^2 q_c \sigma_c^2 + 42L\eta^2 q_{B^2} D_f\left(\mathbf{x},\mathbf{x}^*\right) + 21\eta^2 q_g \sigma_g^2$$

With $7\eta^2\sigma^2 q_B \leq 3 \times \frac{7}{2} \times \frac{1}{35^2} = \frac{21}{2450}$, we bound $\mathbb{E}_t$ as,

$$\mathbb{E}_t \leq \frac{2450}{2429}\left\{7\eta^2\sigma^2 q_B + 14L\eta^2 q_{B^2} D_F\left(\mathbf{x},\mathbf{x}^*\right) + 7\eta^2 q\sigma_c^2 + 21\eta^2 q_g\sigma_c^2 + 42L\eta^2 q_{B^2} D_f\left(\mathbf{x},\mathbf{x}^*\right)\right\}$$

$$\leq \frac{71}{10}\eta^2 q_B\sigma^2 + \frac{71}{5}Lq_{B^2}\eta^2 D_F\left(\mathbf{x},\mathbf{x}^*\right) + \frac{71}{10}q_c\eta^2\sigma_c^2 + 22q_g\eta^2\sigma_g^2 + 43Lq_{B^2}\eta^2 D_f\left(\mathbf{x},\mathbf{x}^*\right)$$

Substitute $\mathbb{E}_t$ into $\mathbb{E}\left[\left\|\mathbf{x}^{t+1} - \mathbf{x}^*\right\|^2\right]$,

$$
\mathbb{E}\left[\left\|\mathbf{x}^{t+1} - \mathbf{x}^*\right\|^2\right]
$$

$$
\leq \left(1 - \frac{\mu SPKR\eta}{2}\right)\left\|\mathbf{x} - \mathbf{x}^*\right\|^2 + 7SPRK\eta^2\sigma^2 + 7S^2R^2P^2K^2\eta^2\frac{(N-P)}{P(N-1)}\sigma_c^2
$$

$$
+ \frac{21P^2K^2R^2S^2(M-S)}{S(M-1)}\eta^2\sigma_g^2 - \frac{8}{5}SPKR\eta D_F\left(\mathbf{x}, \mathbf{x}^*\right)
$$

$$
+ 42LS^2P^2R^2K^2\eta^2 D_f\left(\mathbf{x}, \mathbf{x}^*\right) + \underbrace{\left(2L\eta + 7L^2SPRK\eta^2\right)}_{q_{\mathbb{E}_t} \leq \frac{11}{5}L\eta} \underbrace{\sum_{i,j,r,k}\mathbb{E}\left[\left\|\mathbf{x}_{r,k}^{i,j} - \mathbf{x}\right\|^2\right]}_{\mathbb{E}_t}
$$

$$
\leq \left(1 - \frac{\mu SPKR\eta}{2}\right)\left\|\mathbf{x} - \mathbf{x}^*\right\|^2 + \underbrace{\left[7SPRK\eta^2 + \frac{71}{10}q_B q_{\mathbb{E}_t}\eta^2\right]\sigma^2}_{①}
$$

$$
+ \underbrace{\left[7S^2R^2P^2K^2\frac{(N-P)}{P(N-1)}\eta^2 + \frac{71}{10}q_c q_{\mathbb{E}_t}\eta^2\right]\sigma_c^2}_{②} + \underbrace{\left[\frac{21P^2K^2R^2S^2(M-S)}{S(M-1)}\eta^2 + 22q_g q_{\mathbb{E}_t}\eta^2\right]\sigma_g^2}_{③}
$$

$$
+ \underbrace{\left[q_{D_F} + \frac{71}{5}Lq_{B^2}q_{\mathbb{E}_t}\eta^2\right]D_F\left(\mathbf{x}, \mathbf{x}^*\right)}_{④} + \underbrace{\left[42LS^2P^2R^2K^2\eta^2 + 43Lq_{B^2}q_{\mathbb{E}_t}\eta^2\right]D_f\left(\mathbf{x}, \mathbf{x}^*\right)}_{⑤}
$$

Using $\eta \leq \frac{1}{35LSPRK}$ to simplify above equations,

Bounding ①,

$$
\left[7SPRK\eta^2 + \frac{71}{10}q_B q_{\mathbb{E}_t}\eta^2\right]\sigma^2 \leq \eta^2\sigma^2\left[7SPRK + \frac{71}{10}q_B q_{\mathbb{E}_t}\right] \leq \frac{79}{10}SPRK\eta^2\sigma^2
$$

Bounding ②,

$$
\left[7S^2R^2P^2K^2\frac{(N-P)}{P(N-1)}\eta^2 + \frac{71}{10}q_c q_{\mathbb{E}_t}\eta^2\right]\sigma_c^2
$$

$$
\leq \eta^2\sigma_c^2\left[\frac{7S^2R^2P^2K^2(N-P)}{P(N-1)} + \frac{71}{10}q_c q_{\mathbb{E}_t}\right]
$$

$$
\leq \left[\frac{7S^2R^2P^2K^2(N-P)}{P(N-1)}\eta^2 + \frac{53}{10}SPRLK^3\left(S^2PR^2 + PR^2 + 1\right)\eta^3\right]\sigma_c^2
$$

Bounding ③,

$$
\left[\frac{21P^2K^2R^2S^2(M-S)}{S(M-1)}\eta^2 + 22q_g q_{\mathbb{E}_t}\eta^2\right]\sigma_g^2
$$

$$
= \frac{21P^2K^2R^2S^2(M-S)}{S(M-1)}\eta^2\sigma_g^2 + 22q_g q_{\mathbb{E}_t}\eta^2\sigma_g^2
$$

$$
\leq \left[\frac{21P^2K^2R^2S^2(M-S)}{S(M-1)}\eta^2 + 3P^3K^3R^3S^2L\eta^3\right]\sigma_g^2
$$

Bounding ④:

$$
\left[q_{D_F} + 22Lq_{B^2}q_{\mathbb{E}_t}\eta^2\right]D_F\left(\mathbf{x}, \mathbf{x}^*\right) \leq \left\{-\frac{8}{5}SPKR\eta + 22 \times \frac{11}{2450}SPRK\eta\right\}D_F\left(\mathbf{x}, \mathbf{x}^*\right)
$$

$$
= -\frac{3}{2}SPRK\eta D_F\left(\mathbf{x}, \mathbf{x}^*\right)
$$

Bounding ⑤,

$$\left[42LS^2P^2R^2K^2\eta^2 + 42Lq_{B^2}q_{\mathbb{E}_t}\eta^2\right]D_f\left(\mathbf{x}, \mathbf{x}^*\right)$$

$$\leq 42LS^2P^2R^2K^2\eta^2 + \frac{473}{5}L\eta^2L\eta$$

$$\times \left\{ \begin{array}{l} \frac{R^2P^3K^3S^2(S-1)}{3} + SP^3K^3 \times \frac{R^2(R-1)}{3} + \frac{K^2(K-1)SPR}{3} + \frac{R^2(R-1)S(S-1)P^3K^3}{2} \\ + \frac{R^2P^2K^2S(S-1)(K-1)}{3} + \frac{R(R-1)K^2(K-1)P^2S}{2} \end{array} \right\}$$

$$\leq 45LS^2P^2R^2K^2\eta^2D_f\left(\mathbf{x}, \mathbf{x}^*\right)$$

In this paper, we involve one more edge layer to construct HFL, which is the main difference from HLLi & Lyu (2024), so one more term related to edge server gradient update is generated. Based on our definition $\nabla f_i(\mathbf{x}) := \frac{1}{P}\sum_{j=0}^{P-1}\nabla F_j(\mathbf{x})$, we have following bound for $D_f\left(\mathbf{x}, \mathbf{x}^*\right)$,

$$\mathbb{E}\left[\left\|\sum_{i,j,r,k}\left[\nabla f_i(\mathbf{x}) - \nabla f_i\left(\mathbf{x}^*\right)\right]\right\|^2\right] = \mathbb{E}\left[\left\|\sum_{i,j,r,k}\left[\frac{1}{P}\sum_{j=0}^{P-1}\nabla F_j(\mathbf{x}) - \frac{1}{P}\sum_{j=0}^{P-1}\nabla F_j\left(\mathbf{x}^*\right)\right]\right\|^2\right]$$

$$= \mathbb{E}\left[\left\|\frac{1}{P}\sum_{j=0}^{P-1}\sum_{i,j,r,k}\left[\nabla F_j(\mathbf{x}) - \nabla F_j\left(\mathbf{x}^*\right)\right]\right\|^2\right] \leq \mathbb{E}\left[\left\|\sum_{i,j,r,k}\left[\nabla F_j(\mathbf{x}) - \nabla F_j\left(\mathbf{x}^*\right)\right]\right\|^2\right]$$

Then we have following bound,

$$\mathbb{E}\left[\left\|\mathbf{x}^{t+1} - \mathbf{x}^*\right\|^2\right] \leq \left(1 - \frac{\mu SPKR\eta}{2}\right)\left\|\mathbf{x}^t - \mathbf{x}^*\right\|^2 + \frac{71}{10}SPRK\eta^2\sigma^2$$

$$+ \left[\frac{7S^2R^2P^2K^2(N-P)}{P(N-1)}\eta^2 + \frac{53}{10}SPRLK^3\left(S^2PR^2 + PR^2 + 1\right)\eta^3\right]\sigma_c^2$$

$$+ \left[\frac{21P^2K^2R^2S^2(M-S)}{S(M-1)}\eta^2 + 3P^3K^3R^3S^2L\eta^3\right]\sigma_g^2 - \frac{3}{2}SPRK\eta D_F\left(\mathbf{x}, \mathbf{x}^*\right)$$

$$+ 45LS^2P^2R^2K^2\eta^2D_f\left(\mathbf{x}, \mathbf{x}^*\right)$$

Let $\tilde{\eta} = SPKR\eta$,

$$\mathbb{E}\left[\left\|\mathbf{x}^{t+1} - \mathbf{x}^*\right\|^2\right] \leq \left(1 - \frac{\mu\tilde{\eta}}{2}\right)\left\|\mathbf{x}^t - \mathbf{x}^*\right\|^2 + \frac{79}{10}\frac{\tilde{\eta}^2}{SPKR}\sigma^2$$

$$+ \left[\frac{7\eta^2(N-P)}{P(N-1)} + \frac{53}{10}\tilde{\eta}^3L\left(\frac{1}{P} + \frac{1}{S^2P} + \frac{1}{S^3P^2R^2}\right)\right]\sigma_c^2$$

$$+ \left[\frac{21\tilde{\eta}^2(M-S)}{S(M-1)} + \frac{3L\tilde{\eta}^3}{S}\right]\sigma_g^2 - \frac{3}{10}\tilde{\eta}D_F\left(\mathbf{x}, \mathbf{x}^*\right) \tag{23}$$

Based on Lemma 7 in Li & Lyu (2024), with $a = \frac{\mu}{2}$, $b = \frac{3}{10}$, $S_t = D_F(\mathbf{x}, \mathbf{x}^*)$, $C_1 = \frac{79}{10} \times \frac{\sigma^2}{SPKR} + \frac{7(N-P)}{P(N-1)}\sigma_c^2 + \frac{21(M-S)}{S(M-1)}\sigma_g^2$, $C_2 = \frac{53}{10}\left(\frac{1}{P} + \frac{1}{S^2P} + \frac{1}{S^2P^2R^2}\right)L\sigma_c^2 + \frac{3L}{S}\sigma_g^2$, with $w_t = \left(1 - \frac{\mu\tilde{\eta}}{2}\right)^{-(t+1)}$ we have,

$$\mathbb{E}\left[\left\|\mathbf{x}^{t+1} - \mathbf{x}^*\right\|^2\right] \leq 5\mu\left\|\mathbf{x}^0 - \mathbf{x}^*\right\|^2\exp\left(-\frac{\mu}{2}\tilde{\eta}T\right) + \left\{\frac{27\sigma^2}{SPKR} + \frac{24(N-P)}{P(N-1)}\sigma_c^2 + \frac{70(M-S)}{S(M-1)}\sigma_g^2\right\}\tilde{\eta}$$

$$+ \left\{18L\left(\frac{1}{P} + \frac{1}{S^2P} + \frac{1}{S^2P^2R^2}\right)\sigma_c^2 + \frac{53L}{S}\sigma_g^2\right\} \times \tilde{\eta}^2$$

When $S = M, P = N$ for edge and client fully participation with $D := \left\|\mathbf{x}^0 - \mathbf{x}^*\right\|$, we get,

$$\mathbb{E}\left[f\left(\mathbf{x}^T\right) - f\left(\mathbf{x}^*\right)\right] \leq 5\mu D^2\exp\left(-\frac{\mu\tilde{\eta}T}{2}\right) + \frac{27\tilde{\eta}\sigma^2}{MNRK} + \frac{18L\tilde{\eta}^2(M^2NR^2 + NR^2 + 1)\sigma_c^2}{M^2N^2K^2} + \frac{53L\sigma_g^2\tilde{\eta}^2}{M}$$

By turning the leaving rate carefully, we get:

$$\mathbb{E}\left[f\left(\mathbf{x}^T\right) - f\left(\mathbf{x}^*\right)\right]$$

$$= \tilde{\mathcal{O}}\left(\mu D \exp(-\frac{\mu T}{70L}) + \frac{\sigma^2}{SPKR\mu T} + \frac{(N-P)\sigma_c^2}{\mu TP(N-1)} + \frac{(M-S)}{S(M-1)\mu T}\sigma_g^2 + \frac{L\sigma_g^2}{S\mu^2T^2} + \frac{\left(\frac{1}{P} + \frac{1}{S^2P} + \frac{1}{S^2P^2R^2}\right)L\sigma_c^2}{\mu^2T^2}\right)$$

When $P = N, S = M$ with edge and client fully participation, we have,

$$\mathbb{E}\left[f\left(\mathbf{x}^T\right) - f\left(\mathbf{x}^*\right)\right] = \tilde{\mathcal{O}}\left(\frac{\sigma^2}{\mu MNRKT} + \frac{L(M^2NR^2 + NR^2 + 1)\sigma_c^2}{M^2N^2K^2\mu^2T^2} + \frac{L\sigma_g^2}{M\mu^2T^2} + \mu D^2\exp\left(-\frac{\mu T}{70L}\right)\right)$$

$$(24)$$

$\square$

### C.1.2 GENERAL CONVEX

*Proof.* For general case: When $\mu = 0$, we get following bound based on Eq.23,

$$\mathbb{E}\left[\left\|\mathbf{x}^{t+1} - \mathbf{x}^*\right\|^2\right] \le \left\|\mathbf{x}^t - \mathbf{x}^*\right\|^2 + \frac{79}{10}\frac{\tilde{\eta}^2}{SPKR}\sigma^2$$

$$+ \left[\frac{7\tilde{\eta}^2(N-P)}{P(N-1)} + \frac{53}{10}\tilde{\eta}^3L\left(\frac{1}{P} + \frac{1}{S^2P} + \frac{1}{S^2P^2R^2}\right)\right]\sigma_c^2$$

$$+ \left[\frac{21\tilde{\eta}^2(M-S)}{S(M-1)} + \frac{3L\tilde{\eta}^3}{S}\right]\sigma_g^2 - \frac{3}{10}\tilde{\eta}D_F\left(\mathbf{x}, \mathbf{x}^*\right)$$

Applying Lemma 8 in Li & Lyu (2024) with $b = \frac{3}{10}, \frac{1}{d} = \frac{1}{35L}, \frac{\gamma_0}{\gamma Tb} = \frac{10}{\tilde{\eta}T3}\left\|\mathbf{x}^0 - \mathbf{x}^*\right\|^2, C_1\gamma = \frac{C_1\tilde{\eta}}{b} = \left\{\frac{27\sigma^2}{SPKR} + \frac{24(P-P)}{P(N-1)}\sigma_c^2 + \frac{70(M-S)}{S(M-1)}\sigma_g^2\right\}\tilde{\eta}, C_2\gamma^2 = \frac{C_2\tilde{\eta}}{b} = \left\{18L\left(\frac{1}{P} + \frac{1}{S^2P} + \frac{1}{S^2P^2R^2}\right)\sigma_c^2 + \frac{53L}{S}\sigma_g^2\right\} \times \tilde{\eta}^2$, we get,

$$\mathbb{E}\left[\left\|\mathbf{x}^{t+1} - \mathbf{x}^*\right\|^2\right] \le \frac{10}{\tilde{\eta}T3}\left\|\mathbf{x}^0 - \mathbf{x}^*\right\|^2 + \left\{\frac{27\sigma^2}{SPKR} + \frac{24(N-P)}{P(N-1)}\sigma_c^2 + \frac{70(M-S)}{S(M-1)}\sigma_g^2\right\}\tilde{\eta}$$

$$+ \left\{18L\left(\frac{1}{P} + \frac{1}{S^2P} + \frac{1}{S^2P^2R^2}\right)\sigma_c^2 + \frac{53L}{S}\sigma_g^2\right\} \times \tilde{\eta}^2$$

When $P = N, S = M$ with edge and client fully paticipation with $D := \left\|\mathbf{x}^0 - \mathbf{x}^*\right\|$, we have,

$$\mathbb{E}\left[f\left(\mathbf{x}^T\right) - f\left(\mathbf{x}^*\right)\right] \le \frac{10D^2}{3\tilde{\eta}T} + \frac{27\tilde{\eta}\sigma^2}{MNRK} + \frac{18L\tilde{\eta}^2(M^2NR^2 + NR^2 + 1)\sigma_c^2}{M^2N^2K^2} + \frac{53L\sigma_g^2\tilde{\eta}^2}{M}$$

Then tuning learning rate carefully,

$$2C_1^{\frac{1}{2}}\left(\frac{\gamma_0}{T}\right)^{\frac{1}{2}} = 2\left\{\frac{\sqrt{27}\sigma}{\sqrt{SPRK}} + \sqrt{\frac{RP}{P(N-1)}}\sigma_c + \sqrt{\frac{M-S}{(M-1)S}}\sigma_g\right\} \times \frac{D}{\sqrt{T}}$$

$$= O\left(\frac{\sigma D}{\sqrt{SPRKT}} + \sqrt{\frac{N-P}{P(N-1)T}}D\sigma_c + \sqrt{\frac{M-S}{(M-1)ST}}D\sigma_g\right)$$

$$2C_2^{\frac{1}{3}}\left(\frac{\gamma_0}{T}\right)^{\frac{2}{3}} = 2\left\{18L\left(\frac{1}{P} + \frac{1}{S^2P} + \frac{1}{S^2P^2R^2}\right)\sigma_c^2 + \frac{53L}{M}\sigma_g^2\right\}^{\frac{1}{3}}\frac{D^{\frac{4}{3}}}{T^{\frac{2}{3}}}$$

$$= O\left(\frac{(LD^4\sigma_c^2)^{\frac{1}{3}}}{P^{\frac{1}{3}}T^{\frac{2}{3}}} + \frac{(L\sigma_c^2D^4)^{\frac{1}{3}}}{(ST)^{\frac{2}{3}}P^{\frac{1}{3}}} + \frac{(L\sigma_c^2D^4)^{\frac{1}{3}}}{(SPRT)^{\frac{2}{3}}} + \frac{(L\sigma_g^2D^4)^{\frac{1}{3}}}{M^{\frac{1}{3}}T^{\frac{2}{3}}}\right)$$

$$\frac{d\gamma_0}{T+1} = \frac{35LD^2}{T} = O\left(\frac{LD^2}{T}\right)$$

We have,

$$\mathbb{E}\left[f\left(\mathbf{x}^T\right) - f\left(\mathbf{x}^*\right)\right] = \mathcal{O}\left(\frac{\sigma D}{\sqrt{SPRKT}} + \sqrt{\frac{N-P}{P(N-1)T}}D\sigma_c + \sqrt{\frac{M-S}{(M-1)ST}}\sigma_g\right.$$

$$\left. + \frac{\left(LD^4\sigma_c^2\right)^{\frac{1}{3}}}{P^{\frac{1}{3}}T^{\frac{2}{3}}} + \frac{\left(L\sigma_c^2D^4\right)^{\frac{1}{3}}}{(ST)^{\frac{2}{3}}P^{\frac{1}{3}}} + \frac{\left(L\sigma_c^2D^4\right)^{\frac{1}{3}}}{(SPRT)^{\frac{2}{3}}} + \frac{\left(L\sigma_g^2D^4\right)^{\frac{1}{3}}}{M^{\frac{1}{3}}T^{\frac{2}{3}}} + \frac{LD^2}{T}\right)$$

When $P = N, S = M$ with edge and client fully paticipation, we have,

$$\mathbb{E}\left[f\left(\mathbf{x}^{(T)}\right) - f\left(\mathbf{x}^*\right)\right] = \mathcal{O}\left(\frac{\sigma D}{\sqrt{MNRKT}} + \frac{(L(M^2NR^2 + NR^2 + 1)D^4\sigma_c^2)^{\frac{1}{3}}}{(MNRT)^{2/3}} + \frac{(L\sigma_g^2D^4)^{\frac{1}{3}}}{(MT^2)^{\frac{1}{3}}} + \frac{LD^2}{T}\right)$$

$\square$

### C.1.3 Non-convex Case

**Lemma C.3.** *With Assumption 3.1,3.2,3.3, the learning rate satisfies $\eta \le \frac{1}{35LSKRP}$, the global model updates after one global round should be bounded as follows:*

$$\mathbb{E}\left[F\left(\mathbf{x}^{t+1}\right) - F(\mathbf{x}^t)\right] \le -\frac{SRKP\eta}{2}\mathbb{E}\left[\left\|\nabla F\left(\mathbf{x}^t\right)\right\|^2\right] + L\eta^2 SRKP\sigma^2 + \frac{L^2\eta}{2}\sum_{i,j,r,k}\mathbb{E}\left[\left\|\mathbf{x}_{r,k}^{i,j,t}\mathbf{x}^t\right\|^2\right]$$

$$(25)$$

*Proof.* For CHFL, the model udpates of one global round is,

$$\Delta\mathbf{x} = \mathbf{x}^{t+1} - \mathbf{x}^t = -\eta\sum_{i=1}^{S}\sum_{j=1}^{P}\sum_{r=0}^{R-1}\sum_{k=0}^{K-1}g_{r,k}^{i,j} = -\eta\sum_{i,j,r,k}g_{r,k}^{i,j}$$

where $\mathbf{g}_{r,k}^{i,j} = \nabla F_j(\mathbf{x}_{r,k}^{i,j}, \xi_{r,k}^{i,j})$, thus,

$$\mathbb{E}[\Delta\mathbf{x}] = -\eta\sum_{i,j,r,k}\mathbb{E}\left[\nabla F_j\left(\mathbf{x}_{r,k}^{i,j}\right)\right]$$

We focus on a single training round and drop the superscripts $t$:,

$$\mathbb{E}[F(\mathbf{x} + \Delta\mathbf{x}) - F(\mathbf{x})]$$

$$\le \mathbb{E}[\langle\nabla F(\mathbf{x}), \Delta\mathbf{x}\rangle] + \frac{L}{2}\mathbb{E}\left[\|\Delta\mathbf{x}\|^2\right]$$

$$\le \underbrace{-\eta\sum_{i,j,r,k}\mathbb{E}\left[\left\langle\nabla F(\mathbf{x}), \nabla F_j\left(\mathbf{x}_{r,k}^{i,j}\right)\right\rangle\right]}_{A} + \underbrace{\frac{L\eta^2}{2}\mathbb{E}\left[\left\|\sum_{i,j,r,k}g_{r,k}^{i,j}\right\|^2\right]}_{B}$$

Bounding A,

$$-\eta\sum_{i,j,r,k}\mathbb{E}\left[\left\langle\nabla F(\mathbf{x}), \nabla F_j\left(\mathbf{x}_{r,k}^{i,j}\right)\right\rangle\right] = -\frac{\eta}{2}\sum_{i,j,r,k}\mathbb{E}\left[\|\nabla F(\mathbf{x})\|^2 + \left\|\nabla F_j\left(\mathbf{x}_{r,k}^{i,j}\right)\right\|^2 - \left\|\nabla F_j\left(\mathbf{x}_{r,k}^{i,j}\right) - \nabla F(\mathbf{x})\right\|^2\right]$$

$$\le -\frac{\eta SPRK}{2}\|\nabla F(\mathbf{x})\|^2 - \frac{\eta}{2}\sum_{i,j,r,k}\mathbb{E}\left\|\nabla F_j\left(\mathbf{x}_{r,k}^{i,j}\right)\right\|^2 + \frac{L^2\eta}{2}\sum_{i,j,r,k}\mathbb{E}\left[\left\|\mathbf{x}_{r,k}^{i,j} - \mathbf{x}\right\|^2\right]$$

Bounding B,

$$\frac{L\eta^2}{2}\mathbb{E}\left[\left\|\sum_{i,j,r,k} g_{r,k}^{i,j}\right\|^2\right] \leq \frac{L^2\eta}{2}\mathbb{E}\left[\left\|\sum_{i,j,r,k} g_{r,k}^{i,j} - \sum_{i,j,r,k}\nabla F_j\left(\mathbf{x}_{r,k}^{i,j}\right) + \sum_{i,j,r,k}\nabla F_j\left(\mathbf{x}_{r,k}^{i,j}\right)\right\|^2\right]$$

$$\leq L\eta^2 SPRK\sigma^2 + LSPRK\eta^2\sum_{i,j,r,k}\mathbb{E}\left[\left\|\nabla F_j\left(\mathbf{x}_{r,k}^{i,j}\right)\right\|^2\right]$$

When $\eta \leq \frac{1}{35LSPKR}$, we have $-\frac{\eta}{2}(1 - 2L\eta SKRP)\sum_{i,j,r,k}\mathbb{E}\left[\left\|\nabla F_j\left(\mathbf{x}_{r,k}^{i,j}\right)\right\|\right] < 0$, thus,

$$\mathbb{E}[F(\mathbf{x} + \Delta\mathbf{x}) - F(\mathbf{x})] \leq -\frac{\eta SRKP}{2}\|\nabla F(\mathbf{x})\|^2 + L\eta^2 SRKP\sigma^2 + \frac{L^2\eta}{2}\sum_{i,r,k,j}\mathbb{E}\left[\left\|\mathbf{x}_{r,k}^{i,j} - \mathbf{x}\right\|^2\right]$$

$$-\frac{\eta}{2}\sum_{i,r,k,j}\mathbb{E}\left\|\nabla F_j\left(\mathbf{x}_{r,k}^{t,i}\right)\right\|^2 + L\eta^2 SRKP\sum_{i,r,k,j}\mathbb{E}\left[\left\|\nabla F_j\left(\mathbf{x}_{r,k}^{t,i}\right)\right\|^2\right]$$

$$\leq -\frac{SRKP}{2}\|\nabla F(\mathbf{x})\|^2 + L\eta^2 SRKP\sigma^2 + \frac{L^2\eta}{2}\underbrace{\sum_{i,r,k,j}\mathbb{E}\left[\left\|\mathbf{x}_{r,k}^{i,j} - \mathbf{x}\right\|^2\right]}_{\text{client drift}}$$

**Lemma C.4.** *Let Assumptions 3.1, 3.2,3.4 hold and assume that all the local objectives are $\mu$-strongly convex. If the learning rate satisfies $\eta \leq \frac{1}{35LSPRTK}$, then the client shift can be bounded as:*

$$\mathbb{E}_t \leq 6\eta^2 q_B\sigma^2 + 6\eta^2 q_{B^2}\sigma_c^2 + 6\eta^2 q_{B^2}\sigma_g^2 + 6\eta^2 q_{B^2}\|\nabla f(\mathbf{x})\|^2$$

*where $q_B$ and $q_B^2$ can be found in Eq.21 and Eq.22.*

To bound $\mathbb{E}_t$, we first bound $\mathbb{E}\left[\left\|\mathbf{x}_{r,k}^{i,j} - \mathbf{x}\right\|^2\right]$,

$$\mathbb{E}\left[\left\|\mathbf{x}_{r,k}^{i,j} - \mathbf{x}\right\|^2\right] = \eta^2\mathbb{E}\left[\left\|\underbrace{\sum_{i'=1}^{i}\sum_{r'=0}^{r(i)}\sum_{j'=0}^{j-1}\sum_{k'=0}^{k(i)} g_{r',k'}^{i',j'}}_{B_{i,r,j,k}}\right\|^2\right]$$

$$\leq \eta^2\mathbb{E}\left[\left\|B_{i,j,r,k}\left[g_{r',k'}^{i',j'} - \nabla F_j\left(\mathbf{x}_{r',k'}^{i',j'}\right) + \nabla F_j\left(\mathbf{x}_{r',k'}^{i',j'}\right) - \nabla F_j(\mathbf{x})\right.\right.\right.$$
$$\left.\left.\left.+\nabla F_j(\mathbf{x}) - \nabla f'_i(\mathbf{x}) + \nabla f'_i(\mathbf{x}) - \nabla f(\mathbf{x}) + \nabla f(\mathbf{x})\right]\right\|\right]$$

$$\leq 5\eta^2 B_{i,j,r,k}\mathbb{E}\left[\left\|g_{r',k'}^{i',j'} - \nabla F_j\left(\mathbf{x}_{r',k'}^{i',j'}\right)\right\|^2\right]$$

$$+ 5\eta^2 B_{i,j,r,k}\sum_{i',r',j',k'}\mathbb{E}\left[\left\|\nabla F_j\left(\mathbf{x}_{r',k'}^{i',j'}\right) - \nabla F_j\left(\mathbf{x}\right)\right\|^2\right]$$

$$+ 5\eta^2 B_{i,j,r,k}\sum_{i',r',j',k'}\mathbb{E}\left[\left\|\nabla F_j(\mathbf{x}) - \nabla f_i\left(\mathbf{x}\right)\right\|^2\right]$$

$$+ 5\eta^2 B_{i,j,r,k}\sum_{i',r',j',k'}\mathbb{E}\left[\left\|\nabla f_i\left(\mathbf{x}\right) - \nabla f\left(\mathbf{x}\right)\right\|^2\right] + 5\eta^2 B_{i,j,r,k}^2\|\nabla f\left(\mathbf{x}\right)\|^2$$

Bounding $\mathbb{E}_t$,

$$\mathbb{E}_t \leq 5\eta^2\sigma^2 \sum_{i,j,r,k} B_{i,j,r,k} + 5\eta^2 L^2 \sum_{i,j,r,k} B_{i,j,r,k}^2 \mathbb{E} \left\| \mathbf{x}_{r',k'}^{i',j'} - \mathbf{x} \right\|^2$$

$$+ 5\eta^2\sigma_c^2 \sum_{i,j,r,k} B_{i,j,r,k}^2 + 5\eta^2\sigma_g^2 \sum_{i,j,r,k} B_{i,j,r,k}^2 + 5\eta^2 \sum_{i;j,r,k} B_{i,j,r,k}^2 \|\nabla f(\mathbf{x})\|^2$$

$$\leq 5\eta^2\sigma^2 q_B + 5\eta^2 L^2 q_B \mathbb{E}_t + 5\eta^2\sigma_c^2 q_{B^2} + 5\eta^2\sigma_g^2 q_{B^2} + 5\eta^2 q_{B^2} \|\nabla f(\mathbf{x})\|^2$$

$$\left(1 - 5\eta^2 L^2 q_B\right) \mathbb{E}_t \leq 5\eta^2\sigma^2 q_B + 5\eta^2\sigma_c^2 q_{B^2} + 5\eta^2\sigma_g^2 q_{B^2} + 5\eta^2 q_{B^2} \|\nabla f(\mathbf{x})\|^2$$

With the condition $\eta \leq \frac{1}{35SPRKL}$, we have $5L^2 q_B \eta^2 \leq \frac{1}{490}$, thus,

$$\mathbb{E}_t \leq \frac{490}{489} \times \left\{ 5\eta^2\sigma^2 A + 5\eta^2\sigma_c^2 B + 5\eta^2\sigma_g^2 B + 5\eta^2 B \|\nabla f(\mathbf{x})\|^2 \right\}$$
$$= 6\eta^2 q_B \sigma^2 + 6\eta^2\sigma_c^2 q_{B^2} + 6\eta^2 q_{B^2}\sigma_g^2 + 6\eta^2 q_{B^2} \|\nabla f(\mathbf{x})\|^2$$

Substitute $\mathbb{E}_t$ into $\mathbb{E}\left[F\left(\mathbf{x}^{t+1}\right) - F\left(\mathbf{x}^t\right)\right]$,

$$\mathbb{E}\left[F\left(\mathbf{x}^{t+1}\right) - F\left(\mathbf{x}^t\right)\right] \leq -\frac{SRKP\eta}{2}\|\nabla F\left(\mathbf{x}^t\right)\|^2 + L\eta^2 SRKP\sigma^2 + \frac{L^2\eta}{2} \underbrace{\sum_{i,r,k,j} \mathbb{E}\left[\left\|\mathbf{x}_{r,k}^{i,j} - \mathbf{x}\right\|^2\right]}_{\mathbb{E}_t}$$

$$\leq -\frac{SRKP\eta}{2}\left\|\nabla F\left(\mathbf{x}^t\right)\right\|^2 + \frac{L^{2\eta}}{2} \times 6\eta^2 q_{B^2} \left\|\nabla f\left(\mathbf{x}^t\right)\right\|^2$$

$$+ L\eta^2 SRKP\sigma^2 + \frac{L^2\eta}{2} \times 6\eta^2 q_B \sigma_g^2 + \frac{L^2\eta}{2} \times 6\eta^2 q_{B^2}\sigma_c^2 + \frac{L^2\eta}{2} \times 6\eta^2 q_{B^2}\sigma_g^2$$

$$\leq -\frac{1}{2}SPRK\eta\|\nabla F(\mathbf{x}^t)\|^2 + \left(LSRKP\eta^2 + 3L^2\eta^3 q_B\right)\sigma^2 + 3L^2\eta^3 q_{B^2}\sigma_c^2 + 3L^2\eta^3 q_{B^2}\sigma_g^2$$

Subtracting $F^*$ for both side,

$$\mathbb{E}\left[F\left(\mathbf{x}^{t+1}\right) - F^*\right] \leq \mathbb{E}\left[F\left(\mathbf{x}^t\right) - F^*\right] - \frac{1}{2}\tilde{\eta}\mathbb{E}\left[\left\|\nabla F\left(\mathbf{x}^t\right)\right\|^2\right]$$
$$+ \underbrace{\left(LSRKP\eta^2 + 3L^2\eta^3 q_B\right)}_{\text{①}}\sigma^2 + \underbrace{3L^2\eta^3 q_{B^2}\left(\sigma_c^2 + \sigma_g^2\right)}_{\text{②}}$$

With $\eta \leq \frac{1}{35SPRKL}$, we simplify ①,

$$\left(LSRKP\eta^2 + 3L^2\eta^3 q_B\right)\sigma^2$$
$$= LSRKP\eta_2\sigma^2 \left[1 + 3L\eta \left[\frac{(S-1)RPK}{2} + \frac{(R-1)PK}{2} + \frac{(K-1)}{2}\right]\right]$$
$$\leq LSRKP\eta^2\sigma^2 \left[1 + 3L \times \frac{1}{35SPRKL} \left[\frac{(S-1)RPK}{2} + \frac{(R-1)PK}{2} + \frac{(K-1)}{2}\right]\right]$$
$$\leq LSRKP\eta^2\sigma^2 \left[1 + 3 \times \frac{1}{35} \left[\frac{1}{2} + \frac{1}{2} + \frac{1}{2}\right]\right]$$
$$\leq \frac{6}{5}LSRKP\eta^2\sigma^2$$

Using $\tilde{\eta} = SRKP\eta$, then $\left(LSRKP\eta^2 + 3L^2\eta^3 q_B\right)\sigma^2 \leq \frac{6}{5}\frac{L\tilde{\eta}^2\sigma^2}{8RKP}$. With $\eta \leq \frac{1}{35SPRKL}$, we simplify ②,

$$3L^2\eta^3 q_{B^2}\left(\sigma_c^2 + \sigma_g^2\right)$$

$$= \frac{3L^2 q_{B^2}}{S^3 P^3 K^3 R^3}\left(\sigma_c^2 + \sigma_g^2\right)\eta^3 \leq 3L^2 \times \left\{ \begin{array}{c} \frac{R^3 P^3 K^3 S^2 (S-1)}{3} + \times \frac{SP^3 K^3 R^2 (R-1)}{3} \\ + \frac{K^2 (K-1) SPR}{3} + \frac{R^2 (R-1) S (S-1) P^3 K^3}{3} \\ + \frac{R^2 P^2 K^2 S (S-1)(K-1)}{2} + \frac{R(R-1) K^2 (K-1) P^2 S}{2} \end{array} \right\} \times \frac{\left(\sigma_c^2 + \sigma_g^2\right)}{S^3 P^3 K^3 R^3}$$

$$\leq 3L^2 \times \left\{ \frac{RPK(S-1)}{3} + \frac{R-1}{S3} + \frac{K-1}{3SPR} + \frac{(R-1)(S-1)PK}{2S} \right.$$

$$\left. + \frac{(S-1)(K-1)}{2S} + \frac{(R-1)(K-1)}{2SR} \right\} \times \frac{1}{SPKR} q_\sigma(SP, RK) = \frac{3L^2 q_\sigma\left(S, P, R, K\right)\left(\sigma_g^2 + \sigma_c^2\right)}{SPRK}\tilde{\eta}^3$$

Follow Lemma 8 in Li & Lyu (2024), we have $r_0 = F\left(\mathbf{x}^0\right) - F\left(\mathbf{x}^*\right)$, $\quad \gamma = \tilde{\eta}$, $\quad b = \frac{1}{2}$, $C_1 = \frac{6}{5}\frac{L\sigma^2}{SRKP}$, $\quad C_2 = \frac{3L^2 q_\sigma(S,P,R,K)\left(\sigma_g^2 + \sigma_c^2\right)}{SPRK}$, we have,

$$\min_{0 \leq t \leq T}\left[\left\|\nabla F\left(\mathbf{x}^t\right)\right\|^2\right] \leq \frac{r_0}{b\gamma(T+1)} + \frac{C_1\gamma}{b} + \frac{C_2\gamma^2}{b}$$

$$\leq \frac{2}{\tilde{\eta}T}\left[F\left(\mathbf{x}^0\right) - F\left(\mathbf{x}^*\right)\right] + 2C_1\tilde{\eta} + 2C_2\tilde{\eta}^2$$

$$\leq \frac{2}{\tilde{\eta}T}\left[F\left(\mathbf{x}^0\right) - F\left(\mathbf{x}^*\right)\right] + \frac{12\sigma^2}{5SRKP}\tilde{\eta} + \frac{6L^2 q_\sigma\left(S,P,R,K\right)}{SPRK}\tilde{\eta}^2$$

The convergence rate of non-convex case is followed and $H := f(\mathbf{x}^0) - f(\mathbf{x}^*)$,

$$\min_{0 \leq t \leq T}\mathbb{E}\left[\left\|\nabla F\left(\mathbf{x}^t\right)\right\|^2\right] = \mathcal{O}\left(\frac{\left(L\sigma^2 H\right)^{\frac{1}{2}}}{\sqrt{SRKPT}} + \frac{\left(L^2 H^2 q_\sigma\left(\sigma_g^2 + \sigma_c^2\right)\right)^{\frac{1}{3}}}{\left(SPRKT^2\right)^{\frac{1}{3}}} + \frac{LH}{T}\right)$$

When $P = N, S = M$ with edge and client fully paticipation, we have,

$$\min_{0 \leq t \leq T}\mathbb{E}\left[\left\|\nabla f\left(\mathbf{x}^{(t)}\right)\right\|^2\right] = \mathcal{O}\left(\frac{\left(L\sigma^2 H\right)^{1/2}}{\sqrt{MNRKT}} + \frac{\left(L^2 q_\sigma H^2\right)^{1/3}}{\left(MNRKT^2\right)^{1/3}}\left(\sigma_g^2 + \sigma_c^2\right)^{\frac{1}{3}} + \frac{LH}{T}\right)$$

$\square$

