# OpenReview forum: "ON THE CONVERGENCE OF CYCLIC HIERARCHICAL FEDERATED LEARNING WITH HETEROGENEOUS DATA"
_ICLR.cc/2025/Conference — ICLR 2025 Conference Withdrawn Submission_

### Official Review · Reviewer_P7GH · 2024-10-28

**Soundness:** 2
**Presentation:** 2
**Contribution:** 3
**Rating:** 3
**Confidence:** 3

**Summary:**

This work presents Cyclic Hierarchical Federated Learning (CHFL), a variant of Hierarchical Federated Learning (HFL) that operates in a ring architecture, enabling edge servers to cyclically share model updates. The study aims to provide the first theoretical convergence analysis of CHFL across strongly convex, general convex, and non-convex objectives, demonstrating convergence rates under standard assumptions.

**Strengths:**

The paper aims to make contributions to the theoretical foundations of federated learning, which is commendable. Also, the simulation results are promising.

**Weaknesses:**

The paper suffers from a series of limitations:

1) There are a lot of typos across the text. Below, I highlight just a few for the authors:

** Line 40 --> "When the number of clients (devices) participate in the FL, the communication burden ..." This should be "When the number of clients (devices) participate in the FL increases, the communication burden ..."

** Line 45 --> "this edge-based FL still suffers the performance drop due to the limited" This should be "this edge-based FL still suffers from the performance drop due to the limited num"

** Line 52, it is not clear what "On the other hand" is referring to.

2) There are inconsistencies across the text. For example, the title of the Assumption 3.4 reads as "Intra-Edge & Inter-Edge Data Heterogeneity for Non-Convex objective" However, in the paragraph below it, we have "Assumption 3.4 uses one
weaker assumption to bound the diversity on intra-edge and inter-edge only at the optima for the convex case." So, it is not clear whether the assumption is for convex or non-convex case.

3) The entire premise of the system model of the paper is fundamentally questionable. In particular, this work builds upon the framework of "Cho et al. in Cho et al. (2023)". In that work, the cyclic pattern of client participation (e.g., due to battery limitation is considered). It is really not clear why such limitations exist at the edge servers with abundant resources, which makes naive extension of that framework to hierarchical FL, as done in this paper, questionable.

4) The convergence results are alarming.

** There are two unjustified claims: Line 211 --> "By comparing with the convergence rate of other state-of-the-art HFL algorithms, our convergence rate is the optimal" Line 365 --> "Our convergence rate achieves the optimal rate compared with other FL and HFL". What "optimal" mean in this case? Also, in strong convex case, we can obtain linear (exponential) convergence, which is not demonstrated.

** The more fundamental issue is at line 270, where a lower and an upper bound on the step size is considered. This immediately makes the results questionable. In particular, the result implies that if the step size is too small (e.g., the optimizer moves so slow) we cannot guarantee the convergence, which is counter-intuitive and a very rare outcome. This signals an issue in the convergence analysis of the paper.

**Questions:**

Please refer to the comments above regarding the weaknesses of the paper.

---

### Official Review · Reviewer_WEp5 · 2024-11-02

**Soundness:** 2
**Presentation:** 2
**Contribution:** 2
**Rating:** 3
**Confidence:** 4

**Summary:**

This paper focuses on analyzing the convergence of cyclic hierarchical federated learning (HFL). It integrates the idea of HFL and cyclic FL, and provide theoretical insights into the algorithm for strongly convex, general convex, and non-convex loss functions. The authors also provide intuitions on achieving convergence improvements by clustering clients with different objectives. Experimental results show that the proposed approach can perform better than existing FL, cyclic FL, HFL approaches.

**Strengths:**

1. The paper is generally well written and easy to follow.

2. The authors provide theoretical convergence analysis for cyclic hierarchical federated learning, which has not been studied in existing works.

**Weaknesses:**

1. The convergence analysis for cyclic federated learning and hierarchical federated learning already exist. Given these materials, it is not clear why combining these two analysis is challenging. It is not clear what the technical challenges and the novelties of this paper is, given the previous works.

2. Another important thing is that the motivation of cyclic aggregation is not clear. Intuitively, if the edge server data is non-IID, the model will get biased to the data samples in the edge server coverage after each update. It seems that clustering method is trying to resolve that issue, but during experiments, is the clustering idea also applied to other baselines such as HFL? What is the advantage of applying clustering method to Cyclic HFL compared to applying it to original HFL? I think all these points should become clear.

3. Experiments are conducted using relatively small datasets. It is recommended to confirm the advantage using a larger dataset such as miniImageNet or tinyImageNet.

**Questions:**

Please take a look at the weaknesses above.

---

### Official Review · Reviewer_3hTM · 2024-11-03

**Soundness:** 3
**Presentation:** 3
**Contribution:** 3
**Rating:** 6
**Confidence:** 3

**Summary:**

This work proposes convergence analysis of cyclic hierarchical FL with heterogeneous data under convex, strongly convex and non-convex objectives.  They find that they have the highly desirable speedup effect in terms of both edge server number M and edge round R and prove that clustering clients with similar data will help achieve an optimal convergence improvement if the number of edges is large. The empirical results prove that CHFL can achieve comparable or superior performance in terms of accuracy and convergence speed. The edge training epoch accelerates the convergence speed, and the inter-edge heterogeneity has more effect on convergence speed than the intra-edge heterogeneity in specific conditions.

**Strengths:**

1. This is the first time to propose the convergence analysis of Cyclic HFL and the results are meaningful.
2. The paper is well-organized and easy to read.
3. The experiments are clear and prove the theoretical results.

**Weaknesses:**

1. The analysis results are meaningful. But what are the core technical contributions that drive these conclusions? It is essential for the authors to underscore their technical contributions in a more prominent manner.
2. Line 174 in Assumption 3.3 is for non-convex and Line 178 in Assumption 3.4 is for convex？

**Questions:**

See the weakness above.

---

### Official Review · Reviewer_2xv3 · 2024-11-03

**Soundness:** 2
**Presentation:** 1
**Contribution:** 2
**Rating:** 3
**Confidence:** 4

**Summary:**

The paper presents non-asymptotic convergence results for a cyclic hierarchical federated learning (CHFL) algorithm, which aims to address the scalability and data heterogeneity challenges inherent in federated learning models. The authors provide theoretical analysis under strongly convex, general convex, and non-convex cases. Additionally, the paper claims to optimize clustering policies across edges and clients based on variables like local steps and edge training rounds, aiming for adaptability across diverse network setups. Experiments on standard datasets are included to validate the theoretical findings.

**Strengths:**

1.	The decoupling of the learning rate from data heterogeneity measurement is a notable strength, as it enhances the practicality and adaptability of the algorithm, making it easier to implement.

2.	The paper has a good organization.

3.	The paper is clearly written and easy to read to some extent. However, there are some suggestions for improving the writing below.

**Weaknesses:**

1.	Despite contrasting with existing work in Section 2.2, the paper lacks substantial novelty. The problem setup closely resembles that of Cho et al. (2023) and Li & Lyu (2024), with much of the proof structure based on Li & Lyu. The primary distinctions are minor: for instance, in contrast to Cho et al. (2023), this paper includes R rounds of local updates before moving to the next edge server, whereas Cho et al. does not have local updates. Compared to Li & Lyu (2024), this paper introduces multiple clients per edge server, but this could be interpreted as a syntactic variation of having more mini-batch configurations for a single client. Essentially, these differences do not substantively alter the core problem, making the contribution seem incremental rather than novel.

2.	The paper’s approach may not qualify as "hierarchical federated learning" since true hierarchy typically implies multiple federated learning layers in a star shape. The current setup lacks this characteristic, which could make the term misleading.

3.	The paper claims to test the algorithm on "real-world" datasets, yet the use of standard datasets like MNIST, CIFAR, and Shakespeare might not sufficiently represent real-world scenarios.

4.	The authors claim to provide the best clustering policies by determining the best approach based on the number of edges, clients, local steps, and edge training rounds for various objectives. However, this aspect is underdeveloped. The discussion in Section 4.2 is brief and lacks depth, making it difficult to assess this claim. Furthermore, the clustering policies are not thoroughly supported by experimental validation, which makes the contribution appear premature and lacking in empirical backing.

5.	The paper has several clarity and notation issues, including:
•	Table 1: Unclear definition of "C."
•	Notation inconsistency: "i" is used in "M". (maybe m in M, or i in I!)
•	Inconsistent use of "N" (defined in line 160 but not used in line 162).
•	Inconsistent notation ("all" vs. "\all") in lines 172 and 175.
•	Assumption 3.4, intended for non-convex cases, is many times referenced as convex in lines 192, 214, 215, 268, and 269.
•	Line 214 should refer to "CHFL" instead.
•	Undefined term "mu" in Theorem 4.1, which should clarify if it represents "strongly convex."
•	Line 261: "convex case" should be plural.

6.	The assumption that all edges have the same number of clients is limiting. In Section 4.4, even when subsets are taken, the number of clients remains consistent across edges. It would be insightful to consider the effects of varying client numbers among edges.

7.	Assumption 3.1 restricts variance to be bounded. A more realistic scenario would consider affine variance, as suggested by Ajalloeian et al. (2020) [https://arxiv.org/pdf/2008.00051]. This adjustment could offer a more generalized and applicable framework.

8.	Assumption 3.3 contains an error in defining "Inter-Edge Data Heterogeneity."

9.	The paper repeatedly claims, including in line 211, that “By comparing with the convergence rate of other state-of-the-art HFL algorithms, our convergence rate is the optimal.” However, this usage of "optimal" is misleading, as it does not adhere to the formal definition of optimality in convergence analysis. Without rigorous justification to demonstrate that the convergence rate is indeed the best achievable, this claim appears overstated.

**Questions:**

1.	Given that this setup does not strictly represent hierarchical federated learning, would the authors consider a different terminology to describe their model?

2.	Could the authors elaborate on why standard datasets like MNIST, CIFAR, and Shakespeare were chosen to represent real-world data, and how they justify this choice?

3.	Could the authors clarify the basis for their claim of "optimal" convergence rates? Specifically, what criteria substantiate this claim, and how do they ensure it represents the best achievable rate for this type of algorithm?

4.	How do the authors perceive the novelty of their work in comparison to Cho et al. (2023) and Li & Lyu (2024), given the similarities in the problem setup? In particular, what unique aspects distinguish their approach beyond minor structural adjustments?

5.	How would the algorithm perform if the number of clients varied across edges, rather than being uniform?

6.	Could the authors provide insight into how relaxing the bounded variance assumption to an affine variance assumption (as proposed by Ajalloeian et al. (2020) [https://arxiv.org/pdf/2008.00051]) would impact the theoretical results?

---

### Official Review · Reviewer_9H93 · 2024-11-04

**Soundness:** 2
**Presentation:** 2
**Contribution:** 2
**Rating:** 3
**Confidence:** 4

**Summary:**

This paper introduces Cyclic Hierarchical Federated Learning (CHFL), which incorporates a multi-layer architecture to improve scalability, efficiency, and data heterogeneity handling in federated learning systems. CHFL organizes edge servers in a cyclic (ring) topology, which contrasts with traditional star architectures, allowing edge models to update independently while minimizing communication costs. The authors provide convergence analysis of CHFL across various convex and non-convex objectives.

**Strengths:**

+ For the specific problem and assumptions considered, the analysis is reasonably thorough. It covers three different setups, including strongly convex, convex, and non-convex cases.

+ The presentation quality is reasonable, allowing the reader to easily identify their difference from the existing works and contribution.

**Weaknesses:**

- First, this algorithm adopts a sequential training protocol, where different cells conduct the training one by one. It breaks the general framework of Federated learning (or HFL). Consequently, the time consumption will increase linearly with the number of cells.

- In addition, the previous point makes the analysis simpler than in the standard FL and HFL frameworks, which weakens the contributions of this paper.

- The analysis is built on a very loose bounded data heterogeneity assumption, given in Assumptions 3.3 and 3.4. These constants could be very large. Actually, many works on FL have shown how to get rid of such assumptions, e.g., [R1, R2]. Moreover, a most recent work on HFL [R3] also showed that they can establish convergence guarantee with relying this kind of restrictive assumption.

- In the experiments, it is necessary to include a comparison with the x-axis representing time consumption, as the algorithm sacrifices time to achieve performance gains. Additionally, consider adding a stronger baseline, such as the algorithm proposed in [R3].

[R1]Scaffold: Stochastic controlled averaging for federated learning
[R2] ProxSkip: Yes! Local Gradient Steps Provably Lead to Communication Acceleration! Finally!
[R3] Hierarchical Federated Learning with Multi-Timescale Gradient Correction

**Questions:**

1. Can you overcome the bounded data heterogeneity assumption in your analysis?

2. How does your algorithm compare against baselines in terms of incurred latency?

3. Can you add some better baselines, such as the ones mentioned above?

---

### Note · Authors · 2024-11-22

I have read and agree with the venue's withdrawal policy on behalf of myself and my co-authors.